# Functional Nanomaterials Enhancing Electrochemical Biosensors as Smart Tools for Detecting Infectious Viral Diseases

**DOI:** 10.3390/molecules28093777

**Published:** 2023-04-27

**Authors:** Antonella Curulli

**Affiliations:** Consiglio Nazionale delle Ricerche (CNR), Istituto per lo Studio dei Materiali Nanostrutturati (ISMN), 00161 Rome, Italy; antonella.curulli@cnr.it

**Keywords:** electrochemical (bio)sensors, functional nanomaterials, hybrid nanostructures, immunosensors, aptasensors, genosensors

## Abstract

Electrochemical biosensors are known as analytical tools, guaranteeing rapid and on-site results in medical diagnostics, food safety, environmental protection, and life sciences research. Current research focuses on developing sensors for specific targets and addresses challenges to be solved before their commercialization. These challenges typically include the lowering of the limit of detection, the widening of the linear concentration range, the analysis of real samples in a real environment and the comparison with a standard validation method. Nowadays, functional nanomaterials are designed and applied in electrochemical biosensing to support all these challenges. This review will address the integration of functional nanomaterials in the development of electrochemical biosensors for the rapid diagnosis of viral infections, such as COVID-19, middle east respiratory syndrome (MERS), influenza, hepatitis, human immunodeficiency virus (HIV), and dengue, among others. The role and relevance of the nanomaterial, the type of biosensor, and the electrochemical technique adopted will be discussed. Finally, the critical issues in applying laboratory research to the analysis of real samples, future perspectives, and commercialization aspects of electrochemical biosensors for virus detection will be analyzed.

## 1. Introduction

Viruses are very small and almost invisible infectious agents that replicate within the cells of living organisms and consequently can infect all existing life forms, from animals to plants, including humans and microorganisms such as bacteria. [1]. It should be remembered that the common cold and flu are caused by the action of viruses, but so are more serious diseases such as middle east respiratory syndrome (MERS), influenza, hepatitis, and acquired immune deficiency syndrome (AIDS), among others. Recently, the pandemic induced by coronavirus COVID-19 for its global and rapid spread and for its severe health and economic impact has required the development of effective and low-cost methods for the rapid diagnosis of viral infection. Considering a more general and future scenario, since the viruses are able to mutate quickly and recombine genetic material [2], they can reveal significant resistance and transmissibility, increasing the possibility of a pandemic, especially in a globalized world. Consequently, viruses will probably cause other pandemic events in the future, for which governments, the scientific community and ordinary people need to be more prepared [3,4].

For all these reasons, rapid and accurate diagnosis of a viral infection is essential to provide adequate control of the spread of the disease, allowing the effective isolation of patients, thus limiting an uncontrollable and dangerous transmission.

Currently, the gold-standard techniques applied for virus infection diagnosis are PCR (polymerase chain reaction) and ELISA (enzyme-linked immunosorbent assay) [5]. They provide a proper limit of detection (LOD) and precision, but these analytical methods are laborious and complex, expensive, time-consuming, and require large sample volumes and skilled personnel [6]. Therefore, the development of new and improved strategies able to perform the rapid, simple, and on-site diagnosis of viral diseases at a low cost is mandatory. In this context, biosensors provide rapid and on-site monitoring and real-time determination also in biological fluids [7]. Among various biosensors, electrochemical biosensors have been widely used due to their well-understood biointeraction mechanisms and detection process [8]. Electrochemical biosensors can represent smart detection tools for virus detection as part of an accurate, sensitive, specific, and rapid analysis system [5]. The development of a wide typology of nanomaterials has opened the way for their implementation in the electrochemical sensing and biosensing area for medical diagnostics and environmental and food safety. The introduction of nanomaterials in the design and planning of the electrochemical (bio)sensors has attracted special interest because the nanostructured materials can significantly improve the (bio)sensors’ sensitivity and selectivity, allowing real-time monitoring, involving different synthetic approaches from the conventional to the green ones [9,10,11,12,13,14]. This review examines the options that functional nanomaterials can offer for the development of electrochemical biosensors, in particular for virus detection.

Just a reminder, functional nanomaterials are defined as nanomaterials functionalized by chemical/physical processes for a specific application. Several questions are considered. For example, what kind of nanomaterials are synthesized ad hoc for these electrochemical biosensors? Which peculiar and inherent properties are present; which chemical interactions are available depending on the functionalization? How are they applied in biosensors and in real sample analysis? This review aims to provide an informative overview of which functional nanomaterials are applied to the design and assembly of electrochemical (bio) sensors, highlighting the examples related to the determination of viruses and finally indicating the strengths, limits, and future perspectives.

In the literature, several recent and accurate reviews described virus biosensors in general and the electrochemical ones in particular and compared the conventional analytical methods with those more innovative [5,15,16,17,18,19,20,21,22,23,24,25]. On the other hand, they are focused mainly on the sensing strategies, the role of the electrodic materials and the type of recognition element involved, but no particular emphasis is given to the crucial role of functional nanomaterials in electrochemical biosensors for the detection of viruses.

For this reason, this review is intended to introduce an up-to-date survey of the electrochemical biosensors for virus detection, considering the literature of the last seven years (2017–2023) and evidencing the relevance and the action of functional nanomaterials. The review is articulated by presenting first the different types of nanomaterials used to assemble electrochemical biosensors for the determination of pathogenic viruses. Then in Section 4 are illustrated different examples of how such nanomaterials are used, indicating the different types of sensors included. In this way, the reader can have a clearer idea of the characteristics of nanostructured materials, how they are used, and which types of sensors are involved.

## 2. Electrochemical Biosensors

It is reported in the literature [26] that a biosensor can be described as “an integrated receptor-transducer device, which is capable of providing selective quantitative or semi-quantitative analytical information using a biological recognition element”. In addition, “an electrochemical biosensor is a self-contained integrated device, which is capable of providing specific quantitative or semi-quantitative analytical information using a biological recognition element (biochemical receptor) which is retained in direct spatial contact with an electrochemical transduction element” [26] or alternatively and in a more synthetic way an “electrochemical sensor that has a biological recognition element” [27].

The biorecognition element (BRE) is an enzyme, whole cell, antibody, aptamer, nucleic acid and so on and is coupled to a proper transducing system. The specific interaction between the target molecule and the BRE induces a physicochemical or biological change, turned into a measurable property by the transducer. Considering the electrochemical biosensors, electrochemical signals are generated during biochemical reactions/interactions and are monitored using the most common electroanalytical techniques, as illustrated in Figure 1.

### 2.1. Biorecognition Element and Biosensors Classification

The bioreceptor is the most crucial component of the biosensor device; it is the key to the selectivity, responding only to a particular analyte or molecule of interest, avoiding or at least limiting the interferences from other molecules/compounds present in real complex matrices. Biosensors can be classified as (i) biocatalytic biosensors, including generally an immobilized enzyme for which the recognition and binding of the analyte can induce chemical change(s) and (ii) affinity biosensors, including an immobilized antibody, antigen, DNA, or aptamer for which the binding event does not induce necessarily a chemical reaction.

It is well-known that “the term affinity biosensor refers to a device incorporating immobilized biological receptor molecules that can reversibly detect receptor-ligand interactions with a high differential selectivity and in a non-destructive fashion” [28].

Moreover, affinity biosensors can be identified as immunosensor, aptasensors, genosensors, or cell-based biosensors, according to the different biological recognition elements. Finally, in this review, only examples of affinity biosensors are reported.

In Section 4, significant examples of genosensors, aptasensors and immunosensors for virus detection are illustrated with details regarding probe immobilization methods, detection approaches (label-free, sandwich or competitive), and amplification strategies.

Finally, it must be emphasized that the performance of an electrochemical biosensor is strictly correlated to the immobilization procedure of BREs, thus contributing to the stability and specificity of the biosensor and also preserving the biological activity of the BRE. Common immobilizing strategies include adsorption/physisorption, covalent binding, membrane entrapment, incorporation in gels/hydrogels and or polymers, and combinations thereof, as illustrated in Section 4.

### 2.2. Electrode Materials and Techniques

Many different electrodic materials, from noble metals to conducting polymers, are used for biosensors, being gold, carbon and semiconductors the most widespread.

Gold (Au) is a biocompatible, stable, and conducting material, which is consequently attractive for biosensing applications. The combination and functionalization of gold electrodes with proper molecules, such as, for instance, nanomaterials, can improve its analytical performance [10,29].

Carbon-based electrodes include a variety of materials characterized by the different hybridization of carbon atoms, ranging from graphite to glassy carbon (GC).

GC is a well-known carbon material for application in the biosensing area because it can be easily handled and presents mechanical strength, regenerability, and high conductivity together with a large potential window. On the other hand, the requirement of an accurate pre-treatment procedure before use can be considered a drawback for its application in the electrochemical biosensing area [10].

Semiconductors such as indium tin oxide (ITO) have attracted growing interest as electrode materials because they are cheaper than noble metals, transparent, and easy to handle, and their potential window is larger than that of gold. However, ITO has a significantly lower conductivity [30] and is unstable in acidic environments. [31]. Also, the electron transfer kinetics of ITO are typically worse than those of gold or glassy carbon [32].

Screen-printed electrodes (SPEs) represent low-cost biosensing platforms that have attracted higher relevance than the more conventional ones. In fact, they, combined with electrochemical techniques, can be considered a new generation of miniaturized biosensing devices, encouraging the transition from conventional laboratory equipment to portable devices [33]. It is also to be mentioned that SPEs are produced with different geometries and materials, can be easily modified, and are cost-effective [34,35].

Electrochemical techniques have been considered useful analytical tools characterized by robustness, low cost, easy handling, and the possibility of application to several fields. For all these reasons, they can be preferred to other analytical techniques and approaches.

Different electrochemical techniques can be used, and the most common and mentioned in this review are the following: chronoamperometry (CA), cyclic voltammetry (CV), differential pulse voltammetry (DPV), square wave voltammetry (SWV), and electrochemical impedance spectroscopy (EIS). As will be evident from what will be reported in Section 4, it must be stressed that the most widely used electrochemical techniques are particularly DPV, SWV, and EIS.

As a general comment, the voltammetric techniques measure the current response obtained after the perturbation of an electrode surface using a proper scanning of potential. However, these current signals contain two components, i.e., a faradaic current added to a capacitive one. Only the faradaic portion is proportional to the analyte concentration and can be used to obtain data concerning the target concentrations. When working with concentrations of an analyte at the trace level, faradaic components are frequently much lower than capacitive ones; thus, the interference could become serious. So, pulse techniques such as DPV and SWV have been developed to minimize the capacitive contribution to the total current. This is realized by taking into account the different rates of decay of each contribution with time; in fact, the rate of decay of capacitive currents is faster than faradaic ones in a diffusion-controlled process [36].

A very particular role involves EIS for what concerns the electrochemical biosensing area and, in particular, the electrochemical biosensors for virus determination. EIS is an electrochemical technique that can monitor the interaction between the electrode surface and the analyte following the change in the impedance of the electrode/solution interface. In particular, EIS is a powerful analytical approach for investigating the modifications of the electrode interface properties connected with the biorecognition mechanism. The conventional EIS or Faradaic approach involves molecules undergoing electrochemical reactions, and the corresponding EIS signal is mainly due to impedance changes at the electrode interface, recorded by means of the charge transfer resistance (Rct) [37,38].

For more details about the theories underlying the different electrochemical approaches used for electrochemical biosensors, several books and reviews in the literature are well-known [36,39,40].

## 3. Functional Materials for Electrochemical Sensing

It is well known that nanomaterials and nanostructures have been widely used to design and assemble electrochemical biosensors for application in different fields.

Appropriate functionalization of nanomaterials and nanostructures enhances their relevant physicochemical properties and plays a key role in maximizing and improving the performance of electrochemical biosensors.

In addition, the progress and improvements in the design and synthesis of nanomaterials have allowed the creation of ad hoc tailored nanoscale materials with high surface-to-volume ratio, proper shape, dimensions, and structures, so increasing their integration in the biosensing area.

The following section will include the nanomaterials’ classification and synthesis and evidence of the particular advantages of electrochemical biosensors. It should be highlighted that the main advantage of using nanomaterials to assemble electrochemical biosensors is due to the integration of macroscopic systems such as bulk electrodes with nanostructured materials, so improving the electrical conductivity, the electron transfer from and to the electrode and the electrodic active surface area.

Finally, it should also be evidenced that the terms “nanomaterials” and “nanostructures” in this review are assumed as functional nanomaterials, i.e., as the engineered materials, ad hoc designed and tailored for specific purposes.

In this review, the functional nanomaterials mentioned in the various examples of electrochemical (bio)sensors for the determination of viruses will be briefly introduced. All nanomaterials described in the following subsections and applied to electrochemical biosensors for virus detection are represented in Figure 2.

### 3.1. Zero-Dimensional (0-D) Nanomaterials

A nanomaterial with all its dimension in the nanoscale is classified as a zero-dimensional (0-D) nanomaterial [29].

Gold nanoparticles (AuNPs) are the best-known and widely used 0-D nanomaterial because they are biocompatible, stable and present good conductivity together with a high surface-to-volume ratio. In addition, their synthesis procedures are widespread and easy to perform, including reduction, photochemical reduction, and seed growth [12,41].

Finally, AuNP synthesis with controlled morphology includes a wide range of different nanoparticle geometries, such as spheres, triangles, cubes, stars, and thorns, among others. These non-spherical nanoparticles are assumed as anisotropic nanoparticles, i.e., they have shape-dependent chemical and physical properties and are appealing as well as the spherical ones for electrochemical biosensors applications [42].

The synthesis of nanoparticles of other noble metals, such as Ag or Pt, is carried out with methodologies and approaches similar to those of gold nanoparticles [12].

As a final comment, together with conventional methods of synthesis, green synthesis/biosynthesis methods of nanoparticles have attracted increasing interest because they can limit or avoid the use of toxic solvents and chemicals and the consequent environmental impact. An interesting recent example of green chemistry nanoparticle synthesis is represented by the “phytoengineering” synthesis of hydronium jarosite nanoparticles with a quasi-2D structure involving agro-waste such as avocado seed natural extract [13].

Graphene quantum dots (GQDs), flat 0-D nanomaterials, have gained growing interest because of their peculiar chemical-physical properties for several applications in electro/photo/chemical catalysis, flexible devices, and biosensors, among others.

GQDs are assumed as carbon-based anisotropic nanomaterials with a texturing equivalent to graphene. The morphological properties of GQDs mimic both carbon dots (CDs) and graphene (G). GQDs provide peculiar features such as excellent dispersibility, more abundant active sites than graphene, better tunability in chemical-physical properties, and comparable dimensions to those of the biomolecules [43,44]. The synthetic strategies of GQDs are based on either cutting the larger graphitized carbon materials (top-down) or the fusion of small precursor molecules (bottom-up) [43,44]. In a top-down approach, graphene oxide (GO) or G sheets, CNTs, graphite or carbon fibers are cut at the nanoscale level to obtain 0-D GQDs. In a bottom-up approach, the synthesis of GQDs is realized through a series of chemical reactions involving small molecules as precursors [43,44].

### 3.2. One-Dimensional (1-D) Nanomaterials

A nanomaterial with two dimensions in the nanoscale but with one out of the nanoscale is classified as a one-dimensional (1-D) nanomaterial [29].

Nanotubes, particularly carbon nanotubes (CNTs), represent the best-known 1-D nanomaterials and are widely applied to electrochemical sensors, as reported in the literature [45,46,47,48]. They can be defined as single-walled carbon nanotubes (SWCNTs), double-walled carbon nanotubes (DWCNTs), and multi-walled carbon nanotubes (MWCNTs), depending on the number of graphite layers. Their peculiar properties, such as high conductivity and reactivity, are correlated with their structure, functionality, morphology, and flexibility. CNTs’ chemical functionalization can easily be performed through tubular structure modification [45,46,47,48] for promoting the electron transfer between BREs and the electrodic surface.

Metallic nanotubes and/or nanowires with excellent electrical conductivity are advantageous for assembling electrochemical biosensors, maximizing signal-to-noise ratios because of more efficient mass transport and a lower charging current. It should be stressed that either nanotubes or nanowires can be obtained with different lengths by modulating the experimental parameters [49,50,51].

Considering gold nanotubes (AuNTs), several synthetic approaches have been proposed for short and long AuNTs, such as electrochemical synthesis, seed-mediated synthesis, template method, lithographic methods, and catalytic methods.

Nanofibers typically have diameters in the nanoscale with varying lengths depending on the synthesis process and are considered 1-D nanomaterials [52].

Carbon nanofibers (CNFs), a well-known carbon nanomaterial, provide comparable electrical conductivity and stability to that of both CNTs and conducting polymer nanofibers. The main difference between CNFs and CNTs is related to the stacking of graphene sheets of different shapes, producing more edge sites on the outer wall of CNFs than CNTs, so promoting the electron transfer from and to the analytes. In addition, the surface of CNFs can be activated and further functionalized without damaging their structure. For these reasons, they have attracted growing attention in the biosensing area.

### 3.3. Two-Dimensional (2-D) Nanomaterials

In 2-D nanomaterials (2DMs), two dimensions are not in the nanoscale. 2-D nanomaterials usually involve plate-like shapes, such as nanofilms, nanolayers, and nanocoatings. The electron conductivity is limited through the thickness but delocalized in the plane of the sheet. 2DMs exhibit not only a high surface area-to-volume ratio but also very significant surface reactivity and sensitivity to environmental changes, being the most important key features for applications in the biosensing area. Furthermore, their conductivity and optical properties, together with their unique mechanical characteristics such as durability and flexibility, make these materials particularly appropriate and suitable for assembling innovative and high-performing biosensors. In particular, considering the virus detection, significant selectivity can be achieved by functionalizing 2DMs with antibodies, nucleic acids, proteins, peptides, or aptamers, allowing specific binding to a particular virus or proteins produced by the host organism. Several interesting and recent reviews are available in the literature concerning 2DMs and their applicability in the biosensing area, including virus detection. [53,54,55] It should be emphasized that while interactions between microbes and 2DMs (especially graphene family nanomaterials) have been extensively studied, the interactions of 2DMs with viruses have been much less investigated [55].

Graphene (G) and its derivatives can be considered the most popular 2-D nanomaterials. Graphene is assumed as a two-dimensional honeycomb-like carbon material with a particular basal plane structure, high electronic and thermal conductivity, wide electrochemical potential window and a large surface area, so it is widely used for assembling (bio)sensors for different applications. Several nanomaterials derived from G have been designed and synthesized, and for more details, different reviews are available in the literature [56,57].

I would like to introduce the transition metal dichalcogenides (TMDs), graphene-like 2D nanomaterials inspired by the graphene 2-D structure, including MoS_2_, WS_2_, TaS_2_, etc. TMDs have a sandwich structure where the chalcogen atoms are separated by a plane of metal atoms in two hexagonal planes [58,59]. The atoms are held together through strong covalent bonds, while each thin layer is through rather weak van der Waals forces. Consequently, the layers can be easily separated from each other to form atomic-level sheets. Several methods have been used to synthesize and produce 2D-TMDs nanosheets, which can be classified as top-down and bottom-up methods.

Top-down methods convert bulk crystals and layered compounds into single- or few-layer of 2D-TMDs, using mechanical or liquid exfoliation [60]. Bottom-up approaches involve the growth of layered nanomaterials under proper conditions with suitable atoms or molecules as precursors [60].

Finally, I would like to introduce two interesting classes of 2-D nanomaterials, i.e., metal-organic frameworks (MOFs) and covalent organic frameworks (COFs) [61,62,63].

Metal-organic frameworks (MOFs) can be assumed as porous coordination polymers, synthesized using organic linkers and metal ions or clusters [61]. MOFs are a new type of crystalline material with high surface area and porosity. They are considered flexible structures with scalable sizes, and they have been applied in different fields, including biosensors and sensors, drug delivery, cancer therapy, and catalysis [61]. Various synthetic approaches to MOFs have been reported in the literature, including the coordination modulation method, microwave-assisted synthesis, ultrasound-assisted synthesis, and additive-assisted synthesis [61]. The bioactive molecules could be incorporated into the MOFs’ structures by in-situ addition during the synthesis or by post-synthesis procedures. In addition, MOFs can be considered ideal platforms to prepare nanocomposites using polymers, metal nanoparticles, graphene, carbon nanotubes, and biomolecules because of their tunable sizes with large surface areas and channels of various sizes. The proper integration of MOFs with other functional materials creates multifunctional nanocomposites, not only improving the properties of the starting materials but also providing very peculiar and interesting characteristics. Consequently, MOFs are widely applied in the development of various sensors, especially biosensors, for biomedical applications.

Covalent organic frameworks (COFs) represent a class of porous crystalline materials synthesized starting from organic building blocks via covalent bonds [62,63], resulting in a flexible structure. Their two-dimensional (2D) structures, including high surface areas and accessible cavities or channels with uniform sizes, are considered appealing for many applications, such as drug delivery and (bio)sensors. Alternately, the applications of COFs can be extended through their integration with other functional materials. In addition, the cavities of COF materials have also been used as “nanoreactors for chemical reactions” [62]. In particular, by means of the “confined synthesis” protocol, metal nanoparticles can be prepared within the COFS cavities, with ultrasmall sizes and the so-called “clean surfaces” [62].

### 3.4. Hybrid Nanostructures

Hybrid nanomaterials involve the combination of nanomaterials aiming to improve or develop particular functionalizations not available in the starting nanomaterials. [29,48] Combining different nanomaterials/nanostructure paves the way to assemble and implement high-performance electrochemical biosensors, exploiting and maximizing the peculiarities of the various components of the hybrid nanocomposite. Hybrid nanomaterials have been widely introduced as a transducer, signal amplifier, and/or label in electrochemical biosensors. In the following section, examples of the electrochemical biosensors for virus detection, including different hybrid nanocomposites, will be reported and discussed.

## 4. Viruses Sensing Strategies

Viral contagions represent one of the most important causes of death and global economic damage. For this reason, virus smart detection methods are fundamental to assessing infection spread and circulation. Viruses are not only a real threat to human life, but they can infect plants and animals, so effective methods of analysis are similarly necessary for accurate and prompt environmental control.

Briefly, I would like to introduce some considerations concerning the structure and classification of the viruses.

Viruses can be assumed as intracellular parasites using the genetic material replication system of the host cells; in fact, they do not possess the genetic information necessary for their metabolism and the synthesis of macromolecules such as proteins. They can introduce their genome in the host cells so it can be replicated and reproduced. Virus identification is difficult because of the large number of structures, shapes, genomes, and replication schemes [55].

Viruses include a nucleic acid heart (RNA or DNA) and an outer protein layer called a capsid. The capsid is a single- or double-protein coating involving only one or a few types of structural proteins. It encapsulates the viral genome and preserves it from nucleases. The genome, together with the capsid proteins, constitutes the nucleocapsid. In some virus families, the nucleocapsid is enveloped in a lipid bilayer arising from the modified host cell membrane, covered with an outer layer of glycoproteins. The viral shell includes host proteins and also displays glycosylated trans-membrane proteins as spikes.

Viruses have been classified according to their schemes for the storage and replication of their genomes through DNA and/or RNA intermediates, following the Baltimore classification [64].

The viral particles’ dimensions range from 20 to 400 nm. The viruses can be ranked according to their shape as enveloped, filamentous, icosahedral (or isometric), or head-and-tail viruses.

For example, animal viruses and human immunodeficiency virus (HIV) are considered enveloped viruses with a membrane wrapping the capsid. Other enveloped viruses are avian influenza viruses, SARS-CoV, Ebola virus, Zika virus, MERS-CoV, and SARS-CoV-2.

Icosahedral viruses such as adenoviruses and herpes viruses are considered almost spheres. Filamentous viruses, like plant viruses, are cylindrical and long. Head-and-tail viruses are so called because they present a head, like the icosahedral viruses, containing nucleic acids and a tail similar to the filamentous viruses, and they are able to infect bacteria.

The connection to host cells depends on their family. Enveloped viruses use glycoproteins incorporated in the shell, while non-enveloped viruses use glycoprotein spikes protruding from the capsid to bind to host cells. In the case of head-and-tail viruses, the tail structure enables an effective attachment to host cells.

Analyzing the virus detection approaches, they are based on the direct detection of intact viruses (oldest methods), on the detection of virus molecular fingerprints involving viral proteins and nucleic acids, and on serology through the detection of antibodies as an immunological response of host organisms to viruses [65,66].

I would like to introduce the most common and conventional methods for virus detection. In particular, the conventional plaque and hemagglutination assays allow direct virus determination, but these approaches are time-demanding, require a lot of work in the laboratory, and they are also applicable only to certain types of viruses [55].

The determination of antibodies (serology), produced as an immunological response to the interaction between the virus and the host organism, is a valid alternative to methods based on the direct detection of the virus. This approach allows us to discriminate if an infection can involve the immunoglobulin class G (IgG), a subclass of antibodies remaining for a long time in the host after the infection, or immunoglobulin M (IgM) antibodies, generated just after a viral infection and disappearing rapidly with time.

The most widely used serological tests include enzyme-linked immunosorbent assays (ELISA), immunofluorescence assays, Western blot assays, hemagglutination inhibition, particle agglutination, plaque-reduction neutralization, and complement fixation [55]. It is to be mentioned that viral antigens, especially proteins encoded by a viral genome, can be present in blood if viruses are released after the cell lysis, and they can be determined when viruses are actively replicating. The main approaches for viral antigen detection also include the serological techniques already indicated, such as ELISA, immunofluorescence assays, Western blot assays, but also electrochemiluminescence, radioimmunoassay, and radioimmunobinding assays [65,66].

Nowadays, serology is the gold standard for viral disease diagnosis in biomedical laboratories. Nevertheless, at the beginning of the so-called “window period”, i.e., the time ranging from the first weeks to several months after virus exposure, the quantity of antibodies can result too low. Consequently, the serologic tests seem to be not suitable for the virus’s early detection. Moreover, it is to be underlined that the production of antibodies may not be sufficient in immunodepressed patients. To address these criticalities, nucleic acid-based methods can be considered valid solutions.

Nucleic acids of viruses can be determined just after infection, ignoring the “window period”. Reverse transcription-PCR (RT-PCR) is a nucleic acid method where an enzyme (reverse transcriptase) is used for transforming RNA to its complementary DNA, followed by PCR amplification. However, though this technique is highly sensitive and specific, some limitations have to be evidenced. First of all, it is necessary to isolate a sufficient amount of the viral nucleic acids with a proper purity level for the subsequent amplification step. In fact, the viral genome amount is low considering the total amount of nucleic acid material recovered. So pending a sufficient quantity of viral nucleic acids, there may be delays in diagnosis, increasing the risk of infection, while the virus replicates faster and faster. The PCR process is still costly, time-demanding, and requires skilled personnel. Furthermore, the conventional nucleic acid detection methods imply low sensitivity, high false-negative or false-positive results, and low specificity [66]. Other techniques can deal with the problems of nucleic acid-based systems, such as loop-mediated isothermal amplification (LAMP) processes and rolling circle amplification. Although these methods are more rapid and do not necessitate sophisticated laboratory equipment, they still require skilled personnel.

Given the challenges faced by conventional methods for virus detection, electrochemistry can represent a useful technological approach to overcome these drawbacks. In fact, electrochemical sensing strategies have the potential to achieve rapid, sensitive, selective, easy-to-handle, on-site detection, including also a fast processing time from the sample analysis to the results. In particular, electrochemical (bio)sensors draw particular and increasing attention because they can be easily miniaturized and prepared involving relatively low cost, they ensure fast responses and multiplexed detection options and can include portable (even wearable) equipment. It is well known [7] that in the future, they can have a fundamental role and impact in three clinical strategic areas: point-of-care testing (POCT) for the early detection and regular monitoring of diseases and strictly connected to the virus diffusion control, wearable sensing for continuous monitoring of health and treatment effectiveness, and microphysiological models for the assessment of toxicity and effectiveness of drugs and vaccines during the in vitro phase and for the knowledge of complex diseases.

An appropriate selection of strategies for target recognition and signal transduction has allowed the development of a wide variety of electrochemical assays.

### 4.1. Genosensors

Viruses include a nucleic acid heart (RNA or DNA) and an outer protein layer called the capsid, as already mentioned in the previous section. So a single viral particle can include either an RNA or a DNA genome. A genosensor or DNA biosensor is based on immobilizing a single-stranded oligonucleotide on a transducer surface to identify its complemental DNA sequence through a very specific hybridization, allowing the direct analysis of complex samples [5,67]. Their sensitivity, low limits of detection (LODs), portability, simplicity, fast response time, high sensitivity and selectivity and compatibility with miniaturized detection technologies have justified the wide diffusion of genosensors in the literature. On the other hand, the main disadvantages are represented by relatively higher costs and instrumental complexity, mainly if similar colorimetric devices are considered.

A typical electrochemical genosensor involves an electrode, a capture probe, and a reporter probe. A capture probe, immobilized onto the electrode, recognizes the target analyte, while the reporter probe includes a redox molecule generating an electrochemical signal. Both the capture probe and reporter probe are very specific toward the target DNA. The most common molecules used as probes are, for example, single-stranded oligonucleotides, aptamers or peptides.

The experimental conditions and approaches for probe immobilization are crucial to guarantee a good performance of genosensors, allowing an effective hybridization, also preserving the transducer electrochemical properties and avoiding either a saturation of the sensor surface or a steric hindrance due to an excessive amount of the immobilized probe [5,68]. Among many different probe immobilization strategies, adsorption is the simplest approach, but it is not the most diffused procedure because usually, strong and oriented interactions are preferred for probe immobilization [5] such as covalent bonds and cross-linking. These methods can provide more stable genosensors with better availability of the analyte binding sites.

The electrochemical detection of DNA hybridization involves the changes in electrochemical behavior in the presence or in the absence of complementary DNA targets, and the most diffused methods are label-free and label-based [68]. The label-free approach involves changes in the redox properties of DNA electroactive bases. The electroactive DNA bases underwent a redox process after hybridization, and among the four DNA bases, guanine and adenine were the most electroactive bases. It is to be highlighted that following the hybridization process, the reduction/oxidation peak current of guanine and adenine is lower than that recorded before hybridization.

The major advantage of the label-free method is to provide a simple procedure and rapid hybridization detection. However, a serious problem is represented by the fact that the guanine electrochemical behavior involves high oxidation potential and high background current, probably because of the non-specific adsorption of DNA targets containing guanine bases.

Concerning the label-based method, the introduction of a redox-active indicator, enzyme label or nanoparticles to the DNA sequences or hybridized DNA is involved. On the other hand, in the label-based format, the label can be introduced both on capture and reporter probes [67,68]. If the capture probe is labeled, the analytical response is due to the proximity of the label to the electrode surface and consequently, the electrochemical response can change because the target–probe interaction can modify the distance between the label and the electrode.

Otherwise, if the reporter is labeled, a sandwich-like structure is created as a consequence of the interactions between the target and the capture probe, and then the corresponding electrochemical response is correlated to the concentration of the label itself.

Almost three years ago, the worldwide coronavirus 2019 (CoV-2) pandemic was announced. It is well known that the CoV-2 virus causes Severe Acute Respiratory Coronavirus Syndrome SARS-CoV-2 [69]. This virus results very similar to other coronaviruses such as Bat CoV RaTG13, identified in bat droppings or SARS-CoV, identified in Asian palm civets. Coronaviruses are enveloped viruses with a single-stranded RNA genome, and they can infect not only humans but also animals, including birds and mammals. COVID-19 contains four structural proteins such as spike (S), membrane (M), nucleocapsid (N) and envelope (E) proteins. S protein on the virus surface is responsible for infection transmission.

Generally, COVID-19 transmission can occur via physical contact or airborne droplets, involving symptoms such as cough, tiredness, dyspnea, headache, throat pain, panting and runny nose. The severity of the symptoms depends on the state of health of the infected person and, in particular, on the presence of other pre-existing diseases [69].

As a consequence of the pandemic, a serious crisis in healthcare systems worldwide was evidenced. The most effective method to prevent the spread of SARS-CoV-2 was slowing down the transmission of the virus through the fast and accurate monitoring of the syndrome carriers. Therefore, the diagnosis of COVID-19, being the first step to managing and checking this disease, requires the design and realization of fast, precise, and responsive detection methods [70].

For this reason, as the first example of a genosensor for virus detection, I would like to introduce an electrochemical biosensor based on graphene and supplied with an electrical output system for selective SARS-CoV-2 genetic material detection [71].

The biosensor used gold nanoparticles (AuNPs) coated with antisense oligonucleotides (ssDNA) [72] for detecting viral nucleocapsid phosphoprotein (N-gene) [73]. A paper-based electrochemical platform, based on an Au-microelectrode, included the sensing probes. A simple signal conditioning circuit, integrated with a microcontroller and an algorithm for the computer interface, was employed in the genosensing platform.

Thus, the combination of nanomaterials such as graphene and AuNPs capped with ssDNA has enabled the assembly of an electrochemical biosensing platform for the diagnosis of positive COVID-19 cases. Further, the design of the antisense probes to simultaneously target two regions of the SARS-CoV-N-gene guaranteed the sensor reliability and applicability even if mutation of a region of the viral gene could occur.

Linearity was achieved in a range of RNA concentrations from 585.4 copies/μL to 5.854 × 10^7^ copies/μL.

The genosensor has been applied to samples collected from Vero cells infected with the SARS-CoV-2 virus and clinical samples. The sensor provided an improvement in the electrochemical response only in the presence of the target, with a limit of detection of 6.9 copies/μL. The biosensor was applied to real clinical samples coming from COVID-19-positive subjects and from negative ones with almost 100% accuracy. The results were also validated using the RT-PCR COVID-19 diagnostic kit.

Graphene oxide nanocolloids (GONC) are an electroactive nanomaterial and were used to act at the same time as a transducing platform as well as the electroactive label to assemble a genosensor for the detection of SARS-CoV genomic sequences [74]

GONC [75] can generate an electrochemical signal from the reduction of the electrochemically reducible oxygen functionalities present on their surface, and for this reason, GONC can be included in a biosensing platform for SARS-CoV-2.

The biorecognition element (BRE) consisted of a short-stranded sequence complementary to the RNA-dependent RNA polymerase (RdRp) genome sequences of SARS-CoV-2. Immobilization of the BRE onto the surface of GONC via physical adsorption reduced the number of oxygen-containing groups (OCGs) available for the electrochemical reduction, so the corresponding signal was lower. In the presence of the target, the formation of the probe-target complex affected the non-covalent interactions with the electrode surface, thus producing a partial disconnection of the complex from the probe surface. Consequently, the electroactivity is restored to a certain extent since more OCGs are ready for electrochemical reduction.

Finally, a linear dynamic range of the genosensor was obtained by DPV, ranging from 1 × 10^−10^ to 1 × 10^−5^ mol∙L^−1^ with a LOD of 186 × 10^−9^ mol∙L^−1^. Unfortunately, the sensor selectivity, reproducibility, repeatability, and stability were not investigated, and no data concerning the real samples were provided.

Hepatitis B (HBV) and C (HCV) viruses are widely spread worldwide, and it is appropriate at this point to introduce some considerations on the importance of monitoring and the determination of these two viruses. HBV is transmitted through blood and body fluids, while HCV is only through blood. These two viruses can be developed using contaminated needles, through tattoos and body piercing, through sexual contact, and from mother to baby in childbirth.

HBV belongs to the *Hepadnavirus* family, has a spherical shape [76], and is an enveloped icosahedral DNA virus. It comprises a circular dsDNA genome, a reverse transcriptase (usually called P) and host proteins. In detail, the outer layer is a lipid envelope containing the embedded viral proteins and is denominated as the surface antigen (HBS Ag). These proteins are involved in viral binding and in attacking the host cells. The envelope surrounds an icosahedral nucleocapsid comprised of the core antigen (HBcAg). The nucleocapsid contains the viral nucleic acid and DNA polymerase [77,78].

On the other hand, HCV is a positive-strand RNA virus belonging to the *Flaviviridae* family. The HCV is a small spherical enveloped virion with an icosahedral capsid. The structure consists of an icosahedral lipid membrane with two glycoproteins (called E1 and E2). Its genome also includes non-structural proteins such as NS2, NS3, NS4A, NS4B, NS5A, and NS5B [74,76]. The structural proteins are separated from the non-structural proteins by the short membrane peptide p7 [79].

Due to the ongoing increase in the number of HCV-infected people, the World Health Organization (WHO) has recognized HCV as a principal global health problem.

As the first example, I have to introduce a label-free electrochemical biosensor based on GQDs for detecting HBV-DNA [80]. GQDs were synthesized by means of the fusion of small precursor molecules (bottom-up) [43,44,81] and directly adsorbed onto the GCE surface through van der Waals forces. K_3_[Fe(CN)_6_] was the electrochemical label to detect and monitor the modifications of the electrodic surface [82], and DPV was used to identify and detect such modifications as a result of the DNA capture probe immobilization. The DNA capture probe is complementary to the HBV-DNA as a report probe. Initially, when the DNA capture probe is immobilized onto the GQDs modified electrode surface, the electron transfer from the electrode is hampered. When the HBV-DNA is present in the solution, the DNA capture probe was preferentially bound to HBV-DNA instead of GQDs and the electron transfer from the electrode to K_3_[Fe(CN)_6_] was restored. In particular, a linear concentration range was achieved from 10 nM to 500 nM with a LOD of 1 nM. The selectivity was addressed considering the sensor response in the presence and in the absence of different DNA sequences such as target DNA, single mismatched (SM), and non-complementary (NC). Considering the analytical responses vs. target DNA, SM and NC sequences, a signal decrease was observed from target to SM, while no electrochemical signal was observed in the presence of NC, thus indicating a good selectivity. Unfortunately, the genosensor reproducibility and repeatability were not evaluated. In addition, it was neither applied to real samples nor validated with an external analytical method.

Madurro and co-workers proposed two genosensors for HBV detection, assembling a sensing platform including a single-stranded DNA capture probe specific to HBV, grafted on a gold electrode modified with reduced graphene oxide (electrochemical detection) or gold nanoparticles (optical detection) [83].

After the addition of HBV genomic DNA, an increase in the current peak value was observed. The linear dependence of the electrochemical response on the log HBV-genomic DNA concentration, with a LOD 7.65 pg∙μL^−1^, was indicated, but the linearity range was not reported. The optical assay was performed by using AuNPs, and a shift of the peak wavelength, linearly proportional to the HBV-genomic DNA concentration, with a detection limit of 0.15 ng∙μL^−1^ was evidenced. The selectivity for both the genosensors was tested using HCV as an interfering virus, and the results were promising. Only the optical sensing platform was applied to clinical samples, but recovery data and a comparison with data coming from an external analytical method were not provided. Finally, the two genosensors stability, reproducibility and repeatability were not evaluated.

An impedimetric genosensor was developed for the determination of HCV genotype 1 in human serum based on the hybridization of the capture probe with a complementary target present in the sample [84]. The capture DNA probe was immobilized on the surface of a fluorine-doped tin oxide (FTO) electrode modified with methylene blue (MB) doped silica nanoparticles (MB@SiNPs). They were synthesized using the reverse microemulsion method [85] for wrapping the hydrophilic, polar MB into a negatively charged silica matrix through electrostatic interaction. The silica nanoparticles (SiNPs) served as a signal amplification platform, and MB acted as an electrochemical indicator. FTO was selected as a working electrode because of its high chemical stability, surface area and high capacitive behavior.

EIS has been used as an electrochemical technique because it represents an effective tool for detecting the interaction between the electrode surface and the analyte. Moreover, EIS is a powerful approach for analyzing the interfacial properties related to biorecognition occurring at the electrode surface, as already mentioned in Section 2.2. After the optimization of the experimental conditions, the genosensor showed a dynamic linear range from 100 to 10^6^ copies∙mL^−1^, with a LOD of 90 copies∙mL^−1^. Non-complementary 1 (NC1) and Non-complementary 2 (NC2) DNA sequences were chosen for the selectivity tests, and the results of the EIS investigation evidenced that no hybridization occurred. The genosensor reproducibility was analyzed with acceptable results in terms of RSD% (2.8%). The stability was also tested, and it was noted that the EIS response increased by 15% after 10 days. After 4 weeks, the Rct value increased by 50%, so evidencing a corresponding loss with respect to genosensor initial activity. Unfortunately, the real sample data were not validated with those coming from an external method.

More recently, Madurro and co-workers, previously mentioned for two genosensors for HBV detection [83], developed a genosensor for HCV detection, including gold electrodes modified with graphene oxide functionalized with ethylenediamine (GO-ETD), where the HCV DNA capture probe was immobilized [86]. After the capture probe immobilization onto the sensing platform, the DNA layer acted as a barrier so, hindering electron transfer from and to the electrode. A scheme of the different steps for assembling the genosensor is illustrated in Figure 3.

The genosensor was applied for the detection of the genomic DNA samples of HCV and HBV-positive patients and of genomic RNA samples of Zika virus-positive patients, employing differential pulse voltammetry and K_4_ [Fe (CN)_6_] as an electroactive probe. The genosensor was selective because it was able to distinguish the HCV virus genomic DNA from the HBV genomic DNA and from the Zika virus genomic RNA. Under DPV-optimized experimental conditions, it was observed an inverse linear relationship between the values of the oxidation peak current of the redox probe and the concentration of the samples. The detection limit was 1.36 nmol∙L^−1^ of RNA. Unfortunately, the genosensor stability, reproducibility and repeatability were not evaluated, and the real sample data were not validated with those coming from an external method.

Human papillomaviruses (HPVs) represent a diversified class of dsDNA viruses playing a role in the development of cervical cancer. There are 230 papillomaviruses ranked low, intermediate or high risk considering their role in cervical cancer occurrence [87,88].

The high-risk HPV genotypes are assumed as the main ones responsible for cervical tumor growth, being the third most widespread form of cancer among women in industrialized countries and the second most common cause of death among women in developing countries. Among the 14 known high-risk species of human papillomaviruses, the HPV16 subspecies represents one of the most important and widespread high-risk genotypes. Its detection via cell culture and serological tests is not particularly effective. On the other hand, molecular cancer screening techniques such as hybrid capture assay tests and polymerase chain reaction (PCR) are efficient [5] but are time-demanding and very complex, requiring skilled personnel.

A label-free impedimetric DNA biosensor based on gold nanotubes (AuNTs) has been proposed for the detection of the HPV16 genotype [89].

Due to the peculiar properties of AuNTs for sensing biomolecules, several procedures have been proposed for the synthesis of short and long AuNTs in polycarbonate (PC) templates, including electroless and electrochemical deposition [49,50,51].

The AuNTs-PC template was used as a biosensing platform and was prepared by means of electrodeposition. The EIS measurements and electrochemical responses of the HPV DNA biosensor were investigated. The HPV16 DNA oligonucleotide immobilization and hybridization processes were performed on the AuNTs-PC sensing platform.

After the immobilization process, a resistivity enhancement was evidenced due to electrostatic repulsion among the backbone of the DNA and the negatively-charged phosphate groups. A scheme of the genosensor assembly and HPV detection steps is shown in Figure 4.

It was evidenced that if an external electric field was applied, the amount of DNA immobilized and hybridized was increased, also enhancing the stability and improving the sensor’s analytical performance. An applied electric field can polarize the material by orienting the dipole moments of polar molecules.

Under optimized experimental conditions, a linear concentration range of 0.01 pM–1 mM with a LOD of 1 fM was achieved. The sensor stability (without an electric field) was also investigated, and after 6 weeks of storage at 4 °C, a decrease of only 9% in the electrochemical response was observed. The selectivity was investigated by analyzing the sensor response in the presence and in the absence of target DNA, single mismatched (SM), and non-complementary (NC) DNA sequence, without and with applying an electric field, respectively. Considering the analytical responses vs. target DNA, SM and NC sequences, it is underlined that the resistivity value decreased with the hybridization from the target DNA to NC, evidencing the selectivity of the sensor.

The reproducibility was also analyzed, and the corresponding RSD% (without electric field) were 0.93, 3.77 and 4.76% for NC, SM and target, respectively. On the other hand, the RSD% values for AuNTs-PC with an electric field were 0.88, 3.06 and 3.89% for NC, SM and target, respectively. Unfortunately, real samples were not analyzed.

An electrochemical genosensor based on carbon nanotube/amine-ionic liquid functionalized reduced graphene oxide [NH_2_-IL-rGO/MWCNTs)] nanostructured platform was designed and assembled for HPV16 detection [90]. 3-(2-aminoethyl)-1-propyl-1H-imidazol-3-ium chloride, used as ionic liquid (IL), was synthesized according to the literature [90] and immobilized on GO.

The nanocomposite was prepared by grafting IL onto GO, and it was deposited on a GCE modified with MWCNT and subsequently used for immobilizing aminated DNA probes via covalent bonds using glutaraldehyde (GA) as a cross-linker. In the presence of anthraquinone-2-sulfonic acid monohydrate sodium salt (AQMS) as a redox-active DNA intercalator, the hybridization of aminated DNA probes with the target HPV16 DNA strands (complementary strands) induced an increase in the genosensor response. The strong specific interaction between the immobilized probes and the complementary strands ensured the detection of the HPV16 gene by means of DPV.

Under optimized experimental conditions, a dynamic linear range from 8.5 nM to 10.7 μM with a LOD) of 1.3 nM was obtained. The genosensor repeatability and reproducibility were investigated. Good repeatability and reproducibility results were found with RSD of 2.9% and 5.2%, respectively.

The sensor specificity towards complementary DNA in the presence of a large excess of different DNA sequences was considered. The DNA intercalator response was only obtained in the presence of the complementary DNA, while insignificant responses were observed with the other DNA sequences tested.

To investigate the accuracy and performance of the genosensor, extracted clinical samples DNA were analyzed with recovery data ranging from 94.0 to 102.5%, but these data were not validated with those coming from an external method.

Influenza, an acute, serious and infectious respiratory illness, is worldwide well-known and diffused [91], causing hundreds of thousands of deaths every year, depending on the virus typology. Influenza viruses are enveloped viruses with negative-sense RNA segmented genomes and belong to the *Orthomyxoviridae* family. Influenza viruses are classified as A, B, and C influenza. Influenza A and B viruses are responsible for epidemic influenza (inter-pandemic or seasonal). Influenza A can trigger occasional pandemics, and mild diseases are induced by influenza C. Among these three types of viruses, influenza A is the most violent because it can cause severe and lethal respiratory illnesses. Finally, influenza A and B affect humans, while influenza C generally affects animals [91].

A nanocomposite including binary gold and iron oxide magnetic nanoparticles (Au-MNPs) and CNTs was used to develop a biosensing platform to detect influenza A and norovirus [92]. In particular, the nanocomposite was aligned onto a Pt-interdigitated electrode (Pt-IDE) under a magnetic field, and a DNA probe functionalized with a thiol group was immobilized onto the Au/MNPs-CNT hybrid nanostructure through thiol chemistry. DNA hybridization between the target influenza or norovirus DNA and probe DNA was monitored, evaluating the electrical conductivity change of the nanocomposite, as illustrated in Figure 5.

Under optimized conditions, the analytical performances of the genosensors were investigated by means of LSV, and a linear concentration range from 1 pM to 10 nM was obtained with a LOD for Influenza A and norovirus of 8.4 pM and 8.8 pM, respectively.

The selectivity was analyzed for both the viruses using different mismatched DNA sequences and different viruses such as the Zika virus. The highest response was achieved for the target DNA, evidencing an acceptable selectivity of the genosensor system. Unfortunately, the genosensor stability, reproducibility and repeatability were not evaluated, and real samples were not analyzed.

Cholera is a well-known epidemic induced not by a virus but by a bacterium such as *Vibrio cholerae*, a gram-negative bacterium. Based on the O-antigen classification, more than 200 serogroups were found for this bacterium. In addition, the most effective pathogenic serotypes for human infection were the so-called O1 and O139 [93]. *V. cholerae* is present in aquatic ecosystems and can form biofilm on no-living materials [94]. The *V. cholerae* infections can be triggered by the ingestion of contaminated food or water, and direct transmission between humans is also possible. Generally, the associated disease is accompanied by the following mild symptoms like vomiting and dehydration that, over time, can worsen, causing, in some cases, even the death of the patient. A reliable rapid diagnostics tool represents an important requirement for controlling and preventing cholera infection consequences through early and accurate detection.

An electrochemical genosensor was assembled for the detection of *V. cholerae* using a GCE modified with gold nanocubes and 3-aminopropyltriethoxysilane (APTES) as a sensing platform where a DNA probe was immobilized.

The EIS, CV, Fourier transform infrared spectroscopy (FTIR), and scanning electron microscopy (SEM) techniques were performed to investigate the different steps of the genosensor assembly [95]. The DNA probe was able to identify the nucleotide sequences by means of hybridization. Anthraquinone-2-sulfonic acid monohydrate sodium salt (AQMS) was used as a chemical label. Gold nanocubes acted to improve the electrical conductivity and the electron transfer from and to the electrode surface. After the experimental conditions optimization, the *V. cholerae* detection was performed by means of DPV and two linear, logarithmic concentration ranges were obtained from 1 × 10^−7^ to 1 × 10^−13^ mol∙L^−1^ and from 1 × 10^−13^ to 1 × 10^−27^ mol∙L^−1^, with a LOD of 7.41 × 10^−30^ mol∙L^−1^. The genosensor stability was evaluated over 30 days, and every 7 days, the biosensor response was monitored. The genosensor was stored in a refrigerator at 4 °C. After 1 month of storage, a decrease in the response of 24.68% was determined, and the stability of the biosensor can be considered acceptable. Considering the operational stability tests, it was evidenced that the genosensor can be reused five to six times maximum, without a significant response decrease, even if the percentage of decrease was not reported. The reproducibility was analyzed with an acceptable result in terms of RSD% (1%). The selectivity and specificity tests were carried out in the presence of *S. typhimurium* and *Enterobacter acrogens*, evidencing that the *V. cholerae* electrochemical response was not affected by their presence. The genosensor was applied to spiked real samples of poultry feces with recoveries from 96.42 to 99.11%, but these data were not validated with an external method.

Viral haemorrhagic septicaemia (VHS) is a serious viral infection not for humans but for fish species [96] and is caused by a virus belonging to the *Rhabdoviridae* family. The virus-induced damaging effects on various fish species [97]. It has a similar structure to a bullet with an envelope containing a negative-sense single-stranded RNA genome [98]. Fast and prompt VSH determination in fish farms can be very effective in preventing and/or limiting the virus spread. The conventional methods, including both the serological and the molecular approaches, are time-consuming, expensive, and require skilled personnel.

The first example of an electrochemical genosensor has been reported for the detection of the VHSV Glycoprotein gene, using a PGE modified with a nanocomposite including reduced graphene oxide (rGO) and AuNPs (Au/rGO) [99]. The DNA probe was immobilized onto the modified electrode through thiol chemistry. Different electrochemical techniques such as CV, DPV, and EIS techniques monitored the hybridization, evidencing a decrease in the voltammetric current and an increase in the charge transfer resistance (Rct). In fact, the electron transfer was reduced because of the electrostatic repulsion between the negatively charged DNA and the chemical label K_3_ [Fe (CN)_6_], producing a resistant layer at the surface of the electrode. The electrochemical detection of the VHSV virus was performed by means of EIS, and a linear concentration range was obtained from 1 × 10^−5^ to 10^−10^ mol∙L^−1^ with a LOD of 1.25 × 10^−10^ mol∙L^−1^. The genosensor repeatability and reproducibility were addressed with acceptable results in terms of RSD% (3.42 and 3.70%, respectively). Considering the long-term stability, after 21 days of storage at 4 °C, a decrease of only 15% in the electrochemical response was observed.

For evaluating the biosensor selectivity, its response in the presence of different mismatched sequences was evaluated, and no differences were evidenced in the EIS response with the DNA probe immobilized on the modified PGE.

The applicability of the genosensor to real samples was considered, but the results were not clearly discussed.

As a conclusive comment regarding the reported examples of genosensors, I can observe that the LODs, independently of the analyte, can achieve nM or fM in some examples. On the other hand, a preferred format cannot be evidenced, and selectivity, applicability to real samples, and subsequent validation with an external method, unfortunately, are not generally adequately addressed.

The analytical performances of the nanomaterials involved, together with the sensor format of the reported genosensors for the determination of viruses, are summarized in Table 1.

### 4.2. Immunosensors

Immunosensors have been widely used for detecting different types of viruses, such as avian influenza, SARS-CoV-2, dengue, hepatitis, influenza, and HIV, as reported in this section and summarized in Table 2. Immunosensors have been considered complementary tools to the conventional reverse-transcription polymerase chain reaction (RT-PCR) protocols because they are sensitive and selective, not requiring the PCR sample preparation steps. As already reported in Section 4, the PCR process is still costly, time-demanding, and requires skilled personnel [66].

The working principle of the immunosensing strategy involves the transformation of the results correlated to an immunochemical reaction among antibodies and the corresponding virus target into a measurable signal proportional to the concentration of the analyte. The biorecognition element is an antibody or antigen, usually immobilized on a transducer surface and, in the case of the electrochemical immunosensors, onto the electrode surface [48,100]. The antigens usually are glycoproteins present on the surface of viruses, which are able to bind to the host cell receptor.

On the other hand, the antibodies are specific glycoproteins produced as a response to the antigens’ interaction within the host cell receptor after a few days. Moreover, the presence of antibodies in blood over time represents an indicator of the fact that the patient is/was ill in the past or has been vaccinated [100].

The immobilization approach of the biorecognition element is crucial for the optimal performance of the immunosensor, ensuring the stability of the antibodies or antigens on the electrode and maintaining their specificity and biological activity. Strategies for the immobilization of antibodies are well-known and reported in the literature [101,102]. The adsorption, including electrostatic, hydrophobic, and van der Waals interactions, seems to be attractive but not commonly applied. In fact, it is to be evidenced that this approach results in immobilized randomly oriented antibodies. Consequently, the antigen-binding capacity is reduced, and desorption can occur, limiting the immunosensor stability and reproducibility. On the other hand, covalent immobilization includes the interactions between a functionalized electrode and functionalized antibodies. The immobilization can be performed via a cross-linker such as glutaraldehyde (GA) or via a covalent binding involving 1-ethyl-3-(3-dimethylaminopropyl) carbodiimide (EDC) and *N*-hydroxysuccinimide (NHS, so the sensor stability and reproducibility can be improved.

After the BRE immobilization and incubation of the immunosensor with blocking agents, for instance, bovine serum albumin (BSA), is carried out to prevent non-specific adsorption and avoid a decrease in the sensor sensitivity.

Considering the immunoassay design, electrochemical immunosensors can be classified as label-free, sandwich, and competitive.

A redox probe is present in the solution since antibodies and antigens are not commonly electroactive in the label-free electrochemical immunosensors. The formation of the antibody-antigen immunocomplex reduces and prevents the electron transfer between the electrode and the redox probe.

In sandwich-type format, an immunochemical reaction between the biorecognition element (primary antibody) and the target is involved, followed by the formation of a sandwich complex after the introduction of a labeled secondary antibody. Consequently, an electrochemical signal proportional to the concentration of the analyte is produced. Generally, enzymes and electrocatalysts are used as electroactive labels for the secondary antibody. It must be evidenced the sandwich-format immunosensors showed better analytical performances with respect to those of the label-free ones.

In competitive electrochemical immunosensors, labeled and free biomolecules compete for the binding sites onto the electrodic surface [100]. In this case, immobilization of the antigen is the preferred strategy because of the issues connected with the antibodies’ random orientation. Consequently, the immobilized antigen usually reacts with the labeled antibody in competition with the free antigens involving a corresponding analytical signal decreasing while the free antigen concentration of the sample is increasing.

Avian influenza A viruses (AIVs) are peculiar to wild waterfowl, in particular, the classes of the Anseriformes (ducks and geese) and Charadriiformes (gulls) [103]. AIVs are classified according to the antigenic properties of the surface of the hemagglutinin (HA) and the neuraminidase (NA), their surface glycoproteins. Sixteen subtypes of HA (H1–H16) and nine subtypes of NA (N1–N9) have been identified among wild water birds [100]. AIVs sporadically and periodically can spread from wild birds and infect domestic chickens [104]. After the transmission, these viruses can be further classified as highly pathogenic avian influenza viruses (HPAIVs) showing high pathogenicity in chicken and as low pathogenic avian influenza viruses (LPAIVs) showing low pathogenicity in chicken. Only the LPAIVs of H5 and H7 subtypes can mutate in HPAIVs, inducing severe hemorrhagic disease with mortality rates of 100% and creating a significant threat to the poultry industry [103,104,105,106].

The Zhang group realized a single digital virus electrochemical enzyme-linked immunoassay (digital ELISA) for determining H7N9 avian influenza virus (H7N9 AIV), integrating digital analysis, single molecule electrochemistry (SME) [107,108] enzyme-induced metallization (EIM), as signal amplification method, bifunctional fluorescence magnetic nanospheres (bi-FMNs) as labels and microelectrode array (MA) modified with Au NPs [109]. The modified MA showed a nearly ideal, reproducible electrochemical behavior with narrow redox peaks and small peak separations. A polyclonal antibody (pAb) and alkaline phosphatase (ALP) were coimmobilized onto bi-FMNs. After sandwich immunoreaction, ALP immobilized on bi-FMNs can catalyze the dephosphorylation of *p*-aminophenyl phosphate (*p*-APP), so producing *p*-aminophenol (*p*-AP) and according to the procedure of EIM for electrochemical signal amplification, as already reported in the literature by the same group [110], the virus amount was detected by LSV. A good linear concentration range from 0.01 to 1.5 pg/mL with a LOD of 7.8 fg/mL was achieved. In order to investigate the storage stability of the modified bi-FMNs, the catalytic activity of ALP and biological activity of pAb were monitored for 5 weeks, and the electrochemical response was almost stable. The selectivity was tested by comparing the electrochemical response for H7N9 AIV to those coming from other viruses such as H9N2 AIV, H5N1 AIV, pseudo rabies virus (PRV), and Newcastle disease virus (NDV). The H7N9 AIV response was higher than those of the other viruses. Finally, the H7N9 AIV was tested in complex matrices such as chicken liver or serum, and the corresponding response was not affected by the matrices with respect to that obtained in the buffered solution. Unfortunately, the immunosensor reproducibility and repeatability were not evaluated, and real samples were not analyzed.

A dual-modality immunoassay was developed by Wang and co-workers for H9N2 AIV detection [111], including fluorescent-magnetic-catalytic nanospheres (FMCNs) as labels and alkaline phosphatase (ALP)-induced metallization as a signal amplification strategy, in analogy with the previously mentioned immunosensor for H7N9 AIV detection [106]. FMCNs, Ab, and ALP, were co-immobilized onto the modified ITO electrodes. ITO electrode surface was previously modified with an rGO layer and by means of a stepwise electrodeposition of MnO_2_ and Au nanostructures.

The immunoassay can be applied in real complex samples because of the magnetic properties of FMCNs. Moreover, the fluorescence properties of FMCNs allow the fluorescence method detection, while an amplified electrochemical assay can be obtained through ALP-catalyzed silver deposition. The antibodies on the FMCN surface can act as a target-efficient trap. Consequently, the dual-modality immunoassay merges the advantages of electroanalytical analysis with fluorescence determination so providing an accurate detection device. H9N2 AIV can be detected electrochemically by means of LSV with a linearity range of 0.1–1000 ng∙mL^−1^ and a LOD of 10 pg∙mL^−1^. On the other hand, a linear concentration range from 300 to 1000 ng∙mL^−1^ with an LOD of 69.8 ng∙mL^−1^ was obtained by means of the fluorescence analytical approach.

Other viruses, such as NDV, PRV, H1N1 AIV, and H7N9 AIV, were chosen as interferents for selectivity investigation. They did not affect the electrochemical signal related to H7N9 AIV. Finally, the H9N2 AIV was tested in complex matrices such as fresh chicken serum, chicken lung, chicken liver, and chicken heart and the corresponding response was not affected by the matrices with respect to that obtained in buffered solution.

The repeatability was analyzed, considering the electrochemical response of the six electrodes in parallel using three different H9N2 AIV concentrations, and the obtained RSD was less than 2%. The reproducibility was also evaluated, evidencing an acceptable RSD% (3.2% intra-assay and 6.4% interassay). The stability of FMCNs was investigated by continuously recording the fluorescence data for 7 days, and the corresponding response was quite stable. Unfortunately, the immunosensor was not applied to real samples.

Finally, a label-free format paper-based immunosensor for the detection of different avian influenza virus (H5N1, H7N9, and H9N2) antigens using flexible screen-printed carbon nanotube-polydimethylsiloxane electrodes was described and discussed [112]. The immunosensor assembling involved hydrophobic patterning using screen-printing of the electrodes and drop-casting of single-walled carbon nanotubes functionalized with COOH (COOH-SWCNTs) for the antibody immobilization via EDC/NHS coupling. COOH-SWCNTs were drop-casted onto screen-printed working electrodes, consisting of a paste including multi-walled MWCNTs and polydimethylsiloxane (PDMS). The three AIV (H5N1, H7N9, and H9N2) antibodies were immobilized on three electrodes modified with COOH-SWCNTs, and they were detected using DPV, as shown in Figure 6.

The LODs were 55.7 pg∙mL^−1^ (0.95 pM) for H5N1, 99.6 pg∙mL^−1^ (1.69 pM) for H7N9, and 54.0 pg∙mL^−1^ (0.72 pM) for H9N2, with a common linearity range from 100 pg∙mL^−1^ to 100 ng∙mL^−1^. Different viruses, such as influenza A H1N1 whole viruses, MS2 bacteriophages and H5N1 AIV antigen in PBS and human serum, were tested to investigate the selectivity using a H9N2 AIV immunosensor. Only the electrochemical response for H9N2 AIV decreased, unlike those for the other non-targets indicating a good selectivity. The reproducibility was analyzed with acceptable results in terms of RSD% (3.09%). The immunosensor showed interesting performances towards multiple detections of different viruses, but, unfortunately, data concerning the immunosensor stability and real samples were not provided.

Here, I would like to introduce some interesting examples of immunosensors for the identification and determination of the SARS-CoV-2 virus.

Gao and co-workers developed a multiplexed, portable, wireless electrochemical platform for the detection of SARS-CoV-2 called SARS-CoV-2 RapidPlex [113]. This platform is able to quantitatively detect biomarkers specific to SARS-CoV-2 both in blood and saliva, determining as specific biomarkers: the nucleocapsid protein (NP), the specific immunoglobulins (Igs) against SARS-CoV-2 spike protein (S) (S-IgM and S-IgG), and the C-reactive protein (CRP), within the physiological ranges, using laser engraved graphene (LEG) electrodes. Direct laser writing of graphene electrodes is a very promising technology for the rapid production of two-dimensional carbon materials that can be applied in different sectors ranging from supercapacitors to biosensors, etc. Many carbon-based raw materials can be transformed into graphene by one-step laser scribing, avoiding complex and time-demanding chemical synthesis procedures using different typologies of lasers. Finally, it is to be evidenced that LEG electrodes are assumed as an evolution of the screen-printed electrode (SPE). Several interesting and recent reviews and articles are available in the literature concerning this topic [114,115,116].

SARS-CoV-2 RapidPlex comprises four LEG working electrodes (WEs), a reference electrode (RE), and a graphene counter electrode (CE), all of them patterned on a polyimide (PI) substrate. The determination of the target proteins (NP and CRP) and/or the specific immunoglobulins (S-IgG and S-IgM) is carried out involving a sandwich-format immunoassay. The sandwich-based immunoassays for antigen detection are considered highly sensitive because two different antibodies acting as capture and detector are involved. The required receptors are fixed on the G layer through 1-pyrenebutyric acid (PBA), so avoiding damage to the conjugation of the graphene sheets and improving the stability. In addition, the functional groups of PBA allow a stable immobilization of the capture receptors (specific antibodies or capture proteins) by means of the covalent coupling between their –NH_2_ groups and the carboxylic groups on PBA. The scheme of the wireless Graphene-Based Telemedicine Platform (SARS-CoV-2 RapidPlex) for Rapid and Multiplex Electrochemical Detection of SARS-CoV-2 in blood and saliva is illustrated in Figure 7.

Considering the different targets, the following linear concentration ranges were obtained by means of amperometry: 0.0–500 pg∙mL^−1^ for NP, 0.0–250 ng∙mL^−1^ for SARS-CoV-2 specific IgG and IgM, and 0.0–50 ng∙mL^−1^ for CRP. Reproducibility was analyzed, and the RSD% values, obtained with different biosensors prepared in the same manner on different days, were 6.3%, 8.4%, 6.0%, and 7.6% for CRP, for S1 IgG, for S1-IgM, and the NP antigen, respectively.

Moreover, the selectivity was considered, and biomarkers of similar coronaviruses, including SARS-CoV and MERS-CoV, were tested. No significant cross-reaction for NP, S1-IgG, S1-IgM, and CRP assays in the presence of each tested interferent was evidenced. Finally, the immunosensors showed stable responses over a 5-day storage period at 4 °C for all the targets examined.

To further investigate NP, S1-IgG, S1-IgM, and CRP response to SARS-CoV-2 infection using LEG-based immunosensors, each target molecule was qualitatively and quantitatively determined in serum and saliva samples. The results were comparable to those coming from the same samples analyzed through RT-PCR.

A dual-modality immunosensor based on a screen-printed gold electrode was developed for both colorimetric and electrochemical analysis of SARS-CoV-2 spike antigen [117]. AuNPs were functionalized with a thiol group, and SARS-CoV-2 spike antibodies (mAb) were immobilized on AuNPs through covalent binding via EDC/NHS coupling. Besides the colorimetric determination, the electrochemical determination was carried out with the modified AuNPs in the solution. Since a screen-printed gold electrode was used, a specific interaction occurred between its partially negative charged surface and the partially positive charged AuNPs/mAb surface. Under a voltammetric cathodic scan, the functional groups, such as carbonyl groups, available on the AuNPs/mAb surface can be reduced. In the presence of SARS-CoV-2 spike antigen, antigen-antibody interactions occurred, and the electrochemical response decreased because fewer free groups, such as carbonyl moieties, are found on the surface of mAb. The electrochemical analysis was carried out by means of SWV, and a linear response to the antigen between 1 pg∙mL^−1^ and 10 ng∙mL^−1^ with a detection limit of 1 pg∙mL^−1^ was obtained.

*Streptococcus pneumoniae*, influenza A and MERS-CoV spike antigens were considered to evaluate the selectivity, and no significant response coming from the interferences was evidenced. However, the electrochemical immunosensor reproducibility, repeatability and stability were not evaluated.

The electrochemical determination was performed in spiked saliva samples, and the relative standard deviations and recoveries were from 2.2% to 4.8% and 94.1% to 102.2%, respectively.

A sandwich-type immunosensor was assembled for SARS-CoV-2 spike protein (SP) or nucleocapsid protein (NP) detection using SPE modified with carbon black (CB) as nanomaterial, magnetic beads as immobilizing support and a secondary antibody labeled with alkaline phosphatase [118]. CB supported the sensor sensitivity increase and acted as an electrochemical response amplifier.

The analytical performances of the immunosensor were evaluated by means of DPV, using the standard solution of S and N protein in untreated saliva with a detection limit equal to 19 ng/mL and 8 ng/mL in untreated saliva, respectively, for SP and NP. Considering the linearity range, the calibration curve in saliva for both proteins was described by a non-linear four-parameter logistic calibration plot [118].

Each target protein was qualitatively determined in clinical and saliva samples. The results were comparable to those coming from the nasopharyngeal swabs, analyzed through RT-PCR. However, the electrochemical immunosensor reproducibility, repeatability and stability were not evaluated.

A label-free electrochemical immunosensor for the determination of SARS-CoV-2 S-protein, a biomarker of COVID-19 [119], was assembled using a screen-printed carbon electrode (SPCE) modified with a SiO_2_@UiO-66 core-shell nanocomposite. The nanocomposite included UiO-66 (Universitetet Oslo-66), a Zirconium (IV) carboxylate MOF. The Zr MOF is comprised of inorganic Zr6-octahedra surrounded by 12 terephthalates ligands, providing high surface area and porosity, good thermal conductivity, and chemical stability. However, the inclusion of SiO_2_ nanoparticles into the MOF structure was required for accelerating electron transfer and for increasing the MOF’s low electrical conductivity.

Angiotensin-converting enzyme 2 (ACE2) has been used as a receptor for the S-protein because the connection within the receptor-binding domain of the S protein and the peptidase domain of ACE2 facilitates the SARS-CoV-2 incoming into the host cells. ACE2, a metalloproteinase, is usually employed to mediate COVID-19 viral infection [120]. An illustration of the assembling sensor and the immunosensing mechanism is shown in Figure 8.

The electrochemical analysis was carried out by means of EIS, and a linear concentration range from 100.0 fg∙mL^−1^ to 10.0 ng∙mL^−1^, with a LOD of 100.0 fg∙mL^−1^, was evidenced. Human coronavirus HCOV, l-glucose, l-cysteine, l-arginine, uric acid, dopamine, ascorbic acid, vitamin D, ribavirin, zanamivir, favipiravir, remdesiver, and tenofovir were selected as possible interferents and showed negligible interference effect in the determination of SARS-CoV-2 S-protein, except favipiravir, remdesiver, and tenofovir, which are well-known antiviral drugs. A possible explanation could be linked to the fact that these drugs can block the ACE2 receptor against the S-protein. For their direct interaction with the virus, they can be considered useful for the treatment of the viral infection.

Moreover, the reproducibility and repeatability of the immunosensor were investigated. The relative standard deviation (RSD%) was 4.85%, indicating an acceptable reproducibility.

Considering the potential reusability, the immunosensor can be employed at least 10 times, only rinsing with hydrochloric acid and water to disrupt the binding between the ACE2 and S-protein after each analysis. Finally, the immunosensor was applied to nasal fluid samples, and the satisfactory recovery values ranged from 91.6 to 93.2%. The results were also validated with the PCR test. Unfortunately, immunosensor stability was not considered.

A supersandwich-format electrochemical biosensor based on a nanocomposite containing graphene functionalized with *p*-sulfocalix[8]arene (SCX8) (SCX8-RGO), AuNPs and toluidine blue (TB) (Au@SCX8-RGO-TB) was assembled for SARS-CoV-2 RNA detection [121]. The supersandwich biosensor is constituted of a capture probe (CP), target sequence, label probe (LP), and auxiliary probe (AP) [121,122]. The detection can be obtained by using CP and LP, and long concatemers are produced from the hybridization of AP with LP, so increasing the sensitivity. A concatemer is a long continuous DNA molecule including multiple copies of the same DNA sequence linked in series.

Finally, AuNPs improved the biosensor sensitivity, owing to their good conductivity, large surface area, and adsorption capability. It is to be underlined that calixarenes, such as SCX8, acted as a supramolecular recognition element and supported the action of TB as an electrochemical mediator.

A so-called plug-and-play method was designed to obtain a sensitive, accurate, and rapid detection of SARS-CoV-2 in clinical samples without RNA amplification using an electrochemical immunosensor powered by a smartphone.

After the optimization of the experimental parameters, a good linear relationship using the logarithm of the concentrations was obtained from 10^−17^ to 10^−12^ mol∙L^−1^ with an LOD of 3 amol∙L^−1^. The sensor selectivity and specificity were tested using an artificial one-mismatch target (1 M) and two-mismatch target (2 MT) (selectivity) and SARS-CoV, MERS-COV, HCOV-OC43, influenza A, Epstein-Barr virus, *Mycoplasma pneumoniae*, *Chlamydia pneumoniae*, parainfluenza virus, influenza B virus, adenovirus, *Klebsiella pneumoniae*, Candida albicans, yeast-like fungal spores, and *Legionella pneumophila* (specificity). No significant electrochemical signal was found, suggesting that the immunosensor can be assumed to be both selective and specific. However, reproducibility, repeatability and stability were not addressed.

The sensor was applied to real clinical samples, and the results are comparable to those coming from the real-time reverse transcription PCR (RT-qPCR) method.

Zourob and co-workers designed and prepared an immunosensor for the detection of SARS-CoV-2 virus antigen by immobilizing the virus nucleocapsid (N) protein on carbon nanofiber-modified screen-printed electrodes, functionalized through diazonium salt electrografting [123].

The detection of the virus antigen was performed via swabbing, followed by a competitive analysis using a defined concentration of N protein antibody in the solution. The sample collection and detection steps were integrated into a single platform by coating SPCEs with absorbing cotton filling. SWV technique was used for the detection. The immunosensors were assembled by covering the end of the electrode containing the detection zone with a piece of cotton fiber, avoiding scraping the electrode surface. Finally, they can be used immediately for collecting the nasopharyngeal swabs or stored dry at 4 °C until further use. The cotton was used because of its considerable absorption capacity towards the nasopharyngeal swabs. It is to be underlined that the response of the immunosensor with a cotton tip was almost comparable to that of the uncoated immunosensor. The competition between the immobilized and the free antigen for the free antibody in the solution is realized. Increasing the concentration of free antigens, a decrease in the amount of antibody available to bind to the antigen on the electrode surface is observed, so inducing an increase of the reduction peak current of the ferro/ferricyanide redox label. A linear relationship of the SARS-CoV-2 N antigen concentrations was obtained from 1 to 1000 ng∙mL^−1^ with a LOD of 0.8 pg∙mL^−1^. The immunosensor reproducibility was tested, obtaining acceptable results in terms of the RSD%, ranging from 2.5 to 5.5%. Influenza A and HCOV viruses were selected as possible interferences, and the results indicated a good selectivity of the immunosensor. The sensor was then applied to real clinical samples of nasopharyngeal swabs, with recoveries ranging from 91.0 to 95.5%. Finally, these results were confirmed and validated with RT-PCR.

Liv proposed an electrochemical immunosensing platform for SARS-CoV-2 S-protein antibody detection based on a modified GCE including gold-clusters functionalized with cysteamine, glutaraldehyde, the spike protein of the SARS-CoV-2 antigen and bovine serum albumin [124]. The electrochemical oxidation response of the immunosensor at 0.9V was used to detect the SARS-CoV-2 S-protein antibody. In fact, the groups, such as hydroxyl on the surface of the SARS-CoV-2 S-protein antigen, were oxidized during the anodic scan, and the peak height of the SWV response decreased in the presence of immuno-complex to block the electron transfer. The LOD in buffered solution and in saliva or oropharyngeal swab samples was 0.01 ag∙mL^−1^, with the linearity range from 0.1 to 1000 ag∙mL^−1^. The cross-reactivity of the MERS-coronavirus spike antigen of some enzymes, such as α-amylase and lipase, was considered, and the results indicated a good selectivity of the immunosensor.

The sensor stability was investigated, and a decrease of only 5.9% was recorded after 30 days. Finally, the immunosensor was applied to a spiked real sample of saliva and oropharyngeal swab and the relative standard deviation and recovery values were from 4.99% to 5.74% and 96.97% to 101.99%, respectively.

Liv and co-workers modified this immunosensing platform using EDC and NHS instead of glutaraldehyde [125]. In this case, the LOD was 9.3 ag∙mL^−1^ in a buffered solution with a linearity range from 0.1 fg∙mL^−1^ to 10.0 pg∙mL^−1^. The MERS-CoV spike antigen, influenza A spike antigen and *Streptococcus pneumoniae* antigen were considered for testing the selectivity, and the results indicated a good selectivity of the immunosensor, also in this case.

The sensor stability was investigated, storing it at 4 °C, 25 °C, and 37 °C. After 30 days at 4 °C and 25 °C, the sensor response did not vary appreciably, compared to the first day, while after 30 days at 37°C, a decrease of only 12.4% was recorded. The reproducibility was addressed with acceptable results in terms of RSD% ranging from 1.90% to 3.77% for 1, 10 and 100 fg∙mL^−1^ of spike antibody. Next, the immunosensor was applied to spiked real samples of saliva and oropharyngeal swabs, and the relative standard deviation and recovery values were from 2.34% to 3.16%, and from 96.04% to 97.47%, respectively. These results were compared with those coming from the lateral flow immunoassay (LFIA) method in terms of sensitivity, and the developed biosensor was >10^9^ times more sensitive than the LFIA method. Finally, the analysis results from clinical samples were in good agreement with those obtained for the same samples by means of RT-PCR.

Next, I would like to introduce some examples of immunosensors for the determination of the dengue virus. Dengue hemorrhagic fever is triggered by the dengue virus (DENV), and currently, no vaccine or effective antiviral therapy is available to treat this infection [126]. It is well-known that the infection is transmitted by *Aedes* mosquitoes [127]. DENV belongs to the *Flaviviridae* family, and four serotypes are identified as antigenically distinct and closely correlated, namely, DENV-1, DENV-2, DENV-3, and DENV-4.

All these DENV can induce different typologies of infection, from the asymptomatic one to dengue shock syndrome (DSS).

The dengue virus genome contains three structural proteins (C, M, and E) and seven non-structural (NS) proteins (NS1, NS2a, NS2B, NS3, NS4a, NS4B, and NS5).

NS1 is assumed as the most important non-structural protein involved in viral pathogenesis [126]. The diagnosis of dengue infection is difficult and complex both because it presents non-specific symptoms and because the laboratory analysis techniques are expensive, time demanding and require specialized personnel, so the electrochemical approach can represent a proper alternative because they are sensitive, portable and less expensive than the conventional techniques.

An electrochemical immunosensor was developed using an ITO electrode modified with a nanocomposite containing Langmuir–Blodgett (LB) films of molybdenum disulfide (MoS_2_) and gold nanoparticles (AuNPs) [128]. Further, antibodies specific to dengue NS1 antigen were immobilized onto the nanocomposite through the EDC/NHS coupling. The detection of NS1 antigen was performed by EIS, and a linear concentration range from 10^2^ to 10^8^ ng∙mL^−1^ with a LOD of 1.675 ng∙mL^−1^ was achieved. The immunosensor reproducibility was satisfactory, with an RSD% of 4.3%. Considering the immunosensor long-term stability, after 40 days at 4 °C, the electrochemical response decreased by only 11.4%.

The immunosensor was applied to spiked samples of human serum, and the linear concentration range with the LOD was evaluated in this complex matrix. The linear concentration range was 10^2^−10^7^ ng∙mL^−1,^ and the corresponding LOD was 1.19 ng∙mL^−1^: these results seem to be promising for a possible application to real clinical samples.

A label-free electrochemical immunosensor platform for the detection of dengue virus E-protein (DENV-E protein) was assembled based on a nanocomposite including functionalized AuNPs and N-doped reduced graphene oxide (AuNPs/NSG) [129]. l-Cysteine (l-Cys) acted as a green reducing and stabilizing agent of AuNPs and graphene and provided a suitable functionalized surface for immobilizing the antibody. Under the optimized conditions, a linear concentration range from 0.01–100 ng∙mL^−1^ with a low detection limit of 1.6 pg∙mL^−1^ was obtained using SWV as an electroanalytical technique.

Alpha-fetoprotein (AFP), prostate-specific antigen (PSA), and immunoglobulin G (IgG) were tested as possible interferents, and the changes in the electrochemical responses were less than 5.0% with respect to that from DENV-E protein, indicating a satisfactory selectivity.

The immunosensor reproducibility was satisfactory, with an RSD% of 2.84%. Considering the immunosensor long-term stability, after 16 days at 4 °C in a buffered solution, the electrochemical response decreased by only 8%.

Finally, the sensor was applied to human blood serum and dengue serum samples, and the data were comparable to those coming from the standard ELISA method.

Considering some examples for the detection of HBV, I would like to describe a sandwich-type electrochemical immunosensor for determining HBV surface antigen (HBS Ag); the Hepatitis B surface antigen (HBS Ag) is the specific antigen on the HBV membrane and the first serological marker. It can be determined in blood, and it is considered an important biomarker of HBV [130].

A nanocomposite including GO and AuNPs with good conductivity was used for improving the electron transfer from and to the electrode surface. In addition, a hybrid nanostructured composite with amino-functionalized molybdenum disulfide and cuprous oxide nanoparticles was synthesized to incorporate PtNPs for amplifying the corresponding electrochemical signal and for immobilizing the secondary antibody. Under the optimized experimental conditions, the immunosensor was applied to detect different HBS Ag concentrations by amperometry and a linear relationship between the amperometric data and the logarithmic values of HBS Ag concentration was obtained from 0.5 pg to 200 ng∙mL^−1^, with a LOD of 0.15 pg∙mL^−1^. The reproducibility was investigated with acceptable results in terms of RSD% (2.4%).

Alpha-fetoprotein (AFP), carcinoembryonic antigen (CEA), immunoglobulin G (IgG) and prostate-specific antigen (PSA) were considered for analyzing the immunosensor specificity. The selectivity can be considered acceptable since the electrochemical signal variation in the presence of the interferents was less than 5% of that without interferences.

Considering the immunosensor stability, after four weeks at 4 °C, a decrease of only 10% in the electrochemical response was observed. Finally, the immunosensor was applied to human serum samples, with RSD% ranging from 1.77 to 4.87% and recoveries ranging from 97.8 to 101.7%. The immunosensor real samples data were validated with those coming from the ELISA standard method.

A nanocomposite using AuNPs and rGO was employed to assemble onto SPCEs a label-free immunosensor for the detection of Hepatitis B virus core antigen-antibody (anti-HBcAg), another important serological marker for HBV [131]. The corresponding antigen was immobilized on the modified SPCE using the metal-protein interactions without crosslinkers. Anti-HBcAg was electrochemically detected by means of EIS, and a linearity range from 3.91 ng∙mL^−1^ to 125.00 ng∙mL^−1^ with a LOD of 3.80 ng∙mL^−1^ was achieved. A mixed solution with anti-HBcAg and anti-estradiol at the same concentration was employed to verify the immunosensor selectivity in a complex matrix. The obtained data indicated that the presence of anti-estradiol did not affect the detection of anti-HBS Ag in a real complex matrix. Finally, the immunosensor was applied to real human serum samples with satisfactory results in terms of RSD%, ranging from 1.99 to 6.90%. Unfortunately, the reproducibility, repeatability and stability data were not provided.

An electrochemical immunosensor was developed using a nanocomposite combining polytyramine (PTy) and carbon nanotubes (CNTs) for the detection of another biomarker for HBV, such as core hepatitis B antibody (anti-HBc) [132]. The CNTs carboxyl groups acted for anchoring the antigen (HBcAg) through its amine sites. As a general comment, Pty and CNTs were combined to improve the immobilization and the electron transfer and to limit the fluctuations in the diffusion barrier due to the antigen-antibody interactions. The nanocomposite was characterized by atomic force microscopy and electrochemical techniques. Under optimized experimental conditions, anti-HBc was determined by SWV, and the immunosensor showed a linear response from 1.0 to 5.0 ng∙mL^−1^ and a limit of detection of 0.89 ng∙mL^−1^.

The repeatability and reproducibility were addressed with satisfactory results in terms of RSD%: 6.0 for reproducibility and lower than 1.0% for repeatability.

The specificity of the sensor was studied in blood, a very complex matrix containing several proteins, lipids, cells and electroactive molecules. The obtained data showed that the electrochemical response was not affected in a so complex medium.

Using real blood samples, the immunosensor was able to recognize qualitatively the presence of anti-HBc.

A sandwich-type electrochemical immunosensor was assembled for the detection of HBS Ag, based on Rh core and Pt shell nanodendrites immobilized onto graphene nanosheets functionalized with an amino group (RhPt NDs/NH_2_-GS) as an electrochemical label and AuNPs supported onto polypyrrole nanosheets (Au NPs/PPy NS) as sensing platform [133]. RhPt NDs presented several catalytic sites due to their branched core-shell structure; consequently, RhPt NDs/NH_2_-GS acted as a label and an electrochemical response amplifier.

On the other hand, Au NPs/PPy NS improved the electron transfer to and from the electrode surface, guaranteeing a proper microenvironment for immobilizing the antibodies and thus enhancing the immunosensor analytical performances. After the optimization of the experimental condition, HBS Ag was determined by means of amperometry and a linear concentration range from 0.0005 to 10 ng∙mL^−1^, with a LOD of 166 fg∙mL^−1^, was obtained. The immunosensor reproducibility was considered acceptable in terms of RSD% (3.4%).

CEA, IgG, PSA and BSA were selected as possible interferences for analyzing the immunosensor selectivity. The selectivity can be considered acceptable since the electrochemical signal variation in the presence of the interferents was less than 5% of that without interferences. After storage for 28 days at 4 °C, a decrease in the electrochemical response of only 11.5% was evidenced.

Finally, the biosensor was applied to spiked clinical human serum samples, and the data showed recoveries ranging from 97.6 to 101.3%, with an RSD% of 4.63%.

An electrochemical immunosensor for HCV determination was developed using ZnO nanorods and AuNPs [134]. In detail, the ZnO nanorods were synthesized and deposited onto an AuSPE by microwave hydrothermal procedure [135], and the Au nanoparticles were deposited onto ZnO nanorods by sputtering. HCV-antibody was immobilized onto the nanocomposite through cross-linking using cysteamine and glutaraldehyde. HCV was electrochemically detected by means of CV with a LOD of 0.25 μg∙μL^−1^. However, the data on reproducibility, repeatability and stability were not reported or discussed.

Among recently evolved viral diseases considered as potentially fatal threats, hepatitis E virus (HEV) has been recognized to cause acute hepatitis [136,137,138]. The four principal HEV genotypes (G1–G4) are endemic in many industrialized countries, particularly in Japan and Europe.

A pulse-triggered electrochemical immunosensor was developed using graphene GQDs and AuNPs incorporated in polyaniline nanowires, and then the nanocomposite was deposited on a GCE covered with an electropolymerized polyaniline film [139].

The HEV-antibody was finally immobilized onto the modified GCE via EDC/NHS protocol. HEV was determined by means of EIS, and an external electrical pulse was introduced during the virus accumulation step to improve the immunosensor sensitivity.

In the absence of the virus, a redox peak is observed due to the presence of emeraldine, the most electroactive form salt of polyaniline. However, after the virus addition, a decrease in the current was observed, probably because the presence of the virus induced not only a decrease in the electron transfer from and to the electrode, but also an increase in the solution resistance. A linearity range from 1 fg∙mL^−1^ to 100 pg∙mL^−1^ with a LOD of 0.8 fg∙mL^−1^ was achieved.

Influenza virus A (H1N1 and H3N9), norovirus-like particle (NoV-LP) and Zika virus were chosen as possible interferents, and the biosensor responses to the other viruses are significantly lower. Concerning the immunosensor stability, after 2 weeks at 4°C, no significant decrease in the electrochemical response was evidenced.

The immunosensor was applied to determine HEV in fecal samples. The results are comparable to those coming from the standard RT-qPCR method.

Chen and co-workers proposed two immunosensors for HBS Ag detection based on nanocomposites, including, in both cases, Prussian Blue (PB) as an electrocatalyst.

A label-free immunosensor based on an SPCE modified with a nanocomposite including GO, Fe_3_O_4_ magnetic nanoparticles (MNPs), AuNPs and PB was developed for detecting human hepatitis B surface antigen (HBS Ag) [140]. The nanocomposite acted to guarantee high conductivity and an efficient electron transfer from and to the electrode surface. Finally, AuNPs supported the protein adsorption, so binding HBs Ab to create a sort of AuNPs–Ab nanocomposite.

Under optimized detection conditions, HBS Ag was determined via DPV and a linear concentration range of 0.5 pg∙mL^−1^–200 ng∙mL^−1^ with a LOD of 0.16 pg∙mL^−1^ was obtained.

Human serum albumin (HSA), AFP and CEA were tested as possible interferents and compared with the electrochemical response obtained by HBS Ag only; variations in the electrochemical signal due to the interferents were less than 7%.

Considering the immunosensor stability, after 30 days at 4 °C, the electrochemical response decreased by only 7%. The repeatability was also investigated, and the RSD% of the inter-assay was 2.8%. Finally, the biosensor was applied to spiked clinical serum samples, and the obtained data were comparable to those coming from electrochemiluminescence immunoassay (ECLIA).

A label-free electrochemical immunosensor based on SPCE modified with a hybrid nanomaterial including bimetallic Au@Pt nanoparticles, Prussian blue (PB), and reduced graphene oxide-tetraethylene pentamine nanocomposite (rGO-TEPA) was developed for HBS Ag detection [141], as illustrated in Figure 9. The three-dimensional porous Au@Pt core-shells nanoparticles provided a wide surface area, more active sites, and open structure, also reducing the diffusion resistance, speeding up the electron transfer and increasing the contact surface between the electrode and the analyte. It is to be underlined that the porous Au@Pt nanoparticles supported the immobilization of HBS antibodies (Ab).

The PB-synthesized nanocubes can act not only as electrocatalysts but also improve their stability in a neutral solution of rGO-TEPA nanocomposite. In fact, the PB nanocubes were enveloped in the rGO nanosheets, so creating a stable, ad incastro structure.

Under optimal conditions, the electroanalytical determination was performed by means of DPV. A linear range from 0.25 pg∙mL^−1^ to 400 ng∙mL^−1^ and a low detection limit of 0.08 pg∙mL^−1^ was achieved.

Stability, specificity, and reproducibility were investigated. After 30 days at 4 °C, the electrochemical response decreased by only 5.7% with respect to its initial value. CEA, HSA, AFP, cholesterol (CT), and l-cysteine (l-Cys) were considered interference molecules, and the electrochemical signal was not affected by their presence. Finally, the reproducibility data indicated satisfactory results in terms of RSD% (1.76%).

The immunosensor was applied to spiked clinical samples, with recoveries ranging from 98.16 to 102.53%. The obtained data were comparable to those coming from electrochemiluminescence immunoassay (ECLIA).

Cu-MOF has attracted increasing attention because of its flexible structure, electrical conductivity, and electrocatalytic properties [142].

An electrochemical immunosensor based on a GCE modified with Cu-MOF was designed and assembled for HBS Ag detection [143]. In particular, the Cu-MOF structure was designed starting from copper metal ions and amino terephthalic acid ligands. The Cu-MOF size and morphology were tuned by introducing in the synthetic procedure polyvinyl pyrrolidone (PVP) and triethylamine (TEA) acting as a capping agent.

Finally, amine-functionalized Cu-MOF nanospheres were obtained via a solvothermal synthetic approach. Consequently, a stable immobilization of HBS Ab was performed through a covalent interaction between the carboxyl group of Ab and the amino groups of Cu-MOF via EDC/NHS coupling. In addition, the amine-functionalized Cu-MOF acted as an electrocatalyst and an electrochemical response amplifier. The immunosensor was electrochemically characterized via CV, EIS and DPV. Under optimized experimental conditions, HBS Ag was determined by means of DPV and a linearity range from 1 ng∙mL^−1^ to 500 ng∙mL^−1^ with a detection limit of 730 pg∙mL^−1^ was achieved. Considering the immunosensor reproducibility, the relative standard deviation (RSD) was 3.24%. Moreover, the selectivity was performed by comparing the electrochemical response of the target HBS Ag with those coming from other hepatitis virus biomarkers as interfering molecules, including HAV, HDV and HCV. The HBS Ag electrochemical response was much higher compared with other hepatitis virus markers.

Finally, the biosensor was applied to spiked clinical samples, obtaining recoveries from 79.63 to 92.18%.

Moscone and co-workers developed a sandwich-format Enzyme Linked Immuno Magnetic Electrochemical assay (ELIME) for the detection of the hepatitis A virus (HAV) [144]. It was based on magnetic nanobeads, modified as poly (dopamine), as solid support for the immunocomplex, and an array of 8 SPCEs as a sensing platform. The electrochemical determination was carried out by DPV. The core-shell Fe_3_O_4_@poly (dopamine) magnetic nanoparticles (MNPspDA) were used for supporting the sandwich immunological chain. It is well-known in the literature [48] that MNPs can represent an advantage towards the direct immobilization on the electrode surface or onto larger magnetic beads because they improve the sensitivity of the analysis. A quantitative determination of HAV was obtained with a detection limit of 1 × 10^−11^ IU∙mL^−1^ and a concentration range between 1 × 10^−10^–5 × 10^−7^ IU∙mL^−1^.

The intra-day reproducibility of the analysis was 5% (RSD %).

The specificity of the immunosensor was investigated using Coxsackie B4 (Cox, enterovirus present in sewage and water) as interferent, and the electrochemical response was not significantly affected by the Cox presence. The ELIME assay was applied to spike real samples of tap water with a recovery of 83% (mean value).

Microfluidic paper-based analytical devices (μPADs) represent a promising sensing platform for several application fields ranging from point-of-care diagnosis to food safety [29,145]. The application of paper for analytical devices can be explained by considering, for example, its low cost, flexibility, and easy functionalization with proper groups, as already reported in the literature [48]. In particular, electrochemical μPADs have evidenced peculiar analytical performances, starting from a proper selection of electrode materials, electrochemical technique, and/or recognition elements, as illustrated in recent literature reviews [146,147,148].

I would like to introduce an example of a label-free immunosensing platform for human immunodeficiency virus (HIV) p 24 antigen detection, including an electrochemical μPAD [149].

Briefly, regarding HIV, it is a retrovirus consisting of an envelope protein containing two identical single-stranded RNA molecules. Two main types of HIV have been identified: HIV-1, virulent and prevalent, inducing the worldwide pandemic of acquired immune deficiency syndrome (AIDS) and HIV-2, less dangerous and found mostly in West Africa. No cure or vaccine has been found in the 42 years since 1981, when AIDS was described for the first time, despite considerable studies on the evolution of the disease [150].

Coming back to the immunosensing platform, it included zinc oxide nanowires (ZnO-NWs) directly synthesized on working electrodes (WEs), as shown in Figure 10. Combining EIS as an electrochemical technique with ZnO-NWs can improve the detection of HIV p24 antigen. Moreover, the sensor performance can be enhanced by tuning and modulating the nanomaterial morphologies.

A LOD of 0.4 pg∙ml^−1^ was achieved. However, the data on reproducibility, repeatability and stability were not reported or discussed.

Park and co-workers proposed an immunosensor in sandwich format for the optical and electrochemical detection of influenza A virus, using a nanocomposite including binary gold and iron oxide magnetic nanoparticles (Au-MNPs), carbon nanotubes (AuNP-MNPs-CNTs) and CdSeTeS quantum dots (QDs) [151]. Park and co-workers previously employed the AuNP-MNPs-CNTs nanocomposite for assembling a genosensor for the detection of the same virus, as already reported in Section 4.1 [92].

AuNPs, magnetic nanoparticles (MNPs), and CNTs are mixed into a conductive nanocomposite and support the immobilization of the virus antibodies. Moreover, the MNP served to separate the trapped analyte. Besides, QDs were chosen both as optical and electrochemical signal-generating nanomaterial. In particular, considering the electrochemical detection, the sandwich, including viruses and antibodies, was dissolved in an acidic solution to elute the Cd ions from the QDs. Next, the Cd ions were electrochemically determined by means of DPV. After the optimization of electrochemical parameters, a linear concentration range from 1 fg∙mL^−1^ to 1 μg∙mL^−1^ with a detection limit of 9.05 fg∙mL^−1^ was obtained.

Considering the selectivity, norovirus-like particle (NoV-LP), hepatitis E virus (HEV), Zika virus (ZIKV) and white spot syndrome virus were tested, and both the optical signal and the DPV signal were not affected by the presence of other viruses. The immunosensor stability was investigated, and after 3 weeks at 4 °C, a significant decrease in its response was evidenced, probably due to the progressive degradation of the conjugated antibodies.

Finally, the dual immunosensor was applied in a human serum matrix just to evaluate the matrix effect, but not in spiked real samples. Unfortunately, the data on reproducibility and repeatability were not provided.

H1N1 (swine flu) is a viral infection caused by the influenza A subtype virus [152]. In the serum samples of swine flu patients, the Serum Amyloid A (SAA) protein is present in higher concentration with respect to the healthy subjects, so it can be considered a biomarker for the swine flu diagnosis.

An electrochemical label-free immunosensor was developed for detecting swine flu using a glass electrode covered with an ITO layer. A nanostructured mesoporous carbon (mPC), functionalized with -NH_2_ groups via 3-aminopropyltriethoxysilane (APTES), was deposited onto the electrode surface through an electrophoretic deposition [153]. Next, monoclonal anti-SAA antibody (anti-SAA) was immobilized onto the APTES/mPC/ITO electrode via EDC-NHS covalent chemistry. After the morphological and electrochemical characterization of the modified ITO electrode, the SAA protein detection was performed by means of CV. A linear concentration range from 30 to 70 μg∙mL^−1^ with a LOD of 3.1 μg∙mL^−1^ was obtained. Different biomolecules such as glucose, urea, and annexin were tested as possible interferents because they are present in the serum sample of swine flu patients, and they did not affect the electrochemical response. The immunosensor reproducibility was also addressed with acceptable results in terms of RSD% (2.1%). Finally, the immunosensor was applied in a human serum matrix just to evaluate the matrix effect, but not in spiked real samples. Unfortunately, the data on stability and repeatability were not provided.

The Middle East Respiratory Syndrome Corona Virus (MERS-CoV) is recognized as one of the most pathogenic viruses. Generally, the coronaviruses are classified into four groups, including alpha- (group 1), beta- (group 2), gamma- (group 3) and delta-coronavirus (group 4). MERS-CoV belongs to betacoronavirus group. Its genome encodes 10 proteins. These 10 proteins comprise two replicase polyproteins (ORF1ab and ORF1a), four structural proteins (E, N, S, and M), and four non-structural proteins (ORFs 3, 4a, 4b, and 5) [154,155].

An immunosensor for the determination of MERS-CoV was prepared based on a competitive format using an array of carbon electrodes (DEP) modified with AuNPs. Recombinant spike protein S1 was used as a biomarker for MERS CoV [156]. The electrode array can be modulated for multiple detections of different COVs, and for this reason, the electrochemical determination of HCOV was performed. The biosensor was based on indirect competition between the free virus and the immobilized protein biomarker for a defined concentration of antibody present in the sample. Electrochemical measurements using ferro/ferricyanide as a probe were recorded using SWV. A linear response concentration range from 0.001 to 100 ng∙mL^−1^ and 0.01 to 10,000 ng∙mL^−1^ were observed for MERS-CoV and HCOV, respectively, with a detection limit of 0.4 and 1.0 pg∙mL^−1^ for HCOV and MERS-COV, respectively. FluA and FluB proteins were tested as possible interfering molecules, and the electrochemical responses were not affected by their presence.

The reproducibility and the repeatability data were satisfactory in terms of RSD%: 3.0–6.0% for reproducibility and around 2.0% for repeatability. The sensors were considered stable for around 2 weeks, with only a 2.0% decrease in the electrochemical responses.

The immunosensors were applied to biological fluids and artificial nasal samples spiked with MERS-COV and HCOV antigens, and the corresponding recoveries ranged from 89 to 97% for HCOV and from 95 to 108% for MERS-COV. Finally, these data were comparable to those coming from RT-PCR.

As a conclusive comment regarding the reported examples of immunosensors for virus detection, the LODs achieved pg∙mL^−1^ or ag∙mL^−1^ in many cases. Concerning the immunosensor format, the label-free was preferred, even if a not negligible number of examples equal to 40% of the total make use of the sandwich format.

Once again, I must point out that the analytical data are not, in many cases, sufficiently accurate. As usual, questionable points are also represented by the data relating to the applicability to real samples and subsequent validation with an external method; in fact, these issues are not always adequately addressed, except in very few examples. The analytical performance of the reported immunosensors for the determination of viruses as well as the corresponding sensor formats, are summarized in Table 2.

**Table 2 molecules-28-03777-t002:** Performances of electrochemical immunosensors for the viruses’ detection.

Electrode	Functionalized Nanomaterial	Immunosensor Format	Electrochemical Technique	Analyte/Sample	L. R.	LOD	Recovery %	Reference Method	References
ITO	AuNPs	Sandwich format using MA modified with AuNPs and integrating digital analysis, SME, EIM and bi-FMNs as labels	LSV	H7N9 AIV/-	0.01–1.5 pg∙mL^−1^	7.8 fg∙mL^−1^	-	-	[109]
ITO	FMCNs	Sandwich format using EIM and FMCNs as labels	LSV	H9N2 AIV/-	0.1–1000 ng∙mL^−1^	10 pg∙mL^−1^	-	-	[111]
SPMWCNTE	SWCNTs	Label-free format using SPMWCNTE modified with COOH-SWCNTs	DPV	H5N1 AIV/	100 pg∙mL^−1^–100 ng∙mL^−1^	55.7 pg∙mL^−1^	-	-	[112]
SPMWCNTE	SWCNTs	Label-free format using SPMWCNTE modified with COOH-SWCNTs	DPV	H7N9 AIV/	100 pg∙mL^−1^–100 ng∙mL^−1^	99.6 pg∙mL^−1^	-	-	[112]
SPMWCNTE	SWCNTs	Label-free format using SPMWCNTE modified with COOH-SWCNTs	DPV	H9N2 AIV/	100 pg∙mL^−1^–100 ng∙mL^−1^	54.0 pg∙mL^−1^	-	-	[112]
LEGE	G	Sandwich format using LEGE	Amperometry	SARS-CoV-2 NP/serum, saliva	0.0–500 pg∙mL^−1^	-	-	RT-PCR	[113]
LEGE	G	Sandwich format using LEGE	Amperometry	SARS-CoV-2 S1-IgG and IgM/serum, saliva	0.0–250 ng∙mL^−1^	-	-	RT-PCR	[113]
LEGE	G	Sandwich format using LEGE	Amperometry	SARS-CoV-2 CRP/serum, saliva	0.0–50 ng∙mL^−1^	-	-	RT-PCR	[113]
AuSPE	AuNPs	Label-free format using AuSPE modified with AuNPs	SWV	SARS-CoV-2 SP/saliva	1 pg∙mL^−1^–10 ng∙mL^−1^	1 pg∙mL^−1^	94.1–102.2	-	[117]
SPCE	CB	Sandwich format using SPCE modified with CB	DPV	SARS-CoV-2 SP/saliva	-	19 ng mL^−1^	-	RT-PCR	[118]
SPCE	CB	Sandwich format using SPCE modified with CB	DPV	SARS-CoV-2 NP/saliva	-	8 ng mL^−1^	-	RT-PCR	[118]
SPCE	SiO_2_@UiO-66	Label-free format, including an SPCE modified with SiO_2_@UiO-66 nanocomposite	EIS	SARS-CoV-2 SP/nasal fluid samples	100.0 fg∙mL^−1^–10.0 ng∙mL^−1^	100.0 fg∙mL^−1^	91.6–93.2	PCR	[119]
SPCE	Au@SCX8-rGO-TB	Supersandwich format, including an SPCE modified with Au@SCX8-rGO-TB nanocomposite	DPV	SARS-CoV-2 RNA/clinical samples	10^−17^–10^−12^ mol∙L^−1^	3 amol∙L^−1^	-	RT-PCR	[121]
SPCE	CNFs	Competitive format using an SPCE functionalized with CNFs	SWV	SARS-CoV-2 NP/	1–1000 ng∙mL^−1^	0.8 pg∙mL^−1^	91–95.5	RT-PCR	[123]
GCE	AuNCs	Label-free-format using a GCE modified with AuNCs	SWV	SARS-CoV-2 ab/saliva, oropharyngeal swab samples	0.1–1000 ag∙mL^−1^	0.01 ag∙mL^−1^	96.97–101.99	-	[124]
GCE	AuNCs	Label-free format using a GCE modified with AuNCs	SWV	SARS-CoV-2 ab/saliva, oropharyngeal swab samples	0.1 fg∙mL^−1^–10.0 pg∙mL^−1^	9.3 ag∙mL^−1^	96.04–97.47	LFIA	[125]
ITOE	MoS_2_ nanosheets and AuNPs	Label-free format using an ITOE modified with MoS_2_ nanosheets and AuNPs	EIS	DENV NS1P/human serum	10^2^–10^8^ ng∙mL^−1^	1.675 ng∙mL^−1^	-	-	[128]
GCE	AuNPs/NSG	Label-free format using a GCE modified with AuNPs/NSG nanocomposite	SWV	DENV E-protein/human serum clinical samples	0.01–100 ng∙mL^−1^	1.6 pg∙mL^−1^	-	ELISA	[129]
GCE	AuNPs, rGO, PtNPs and MoS_2_@Cu_2_O	Sandwich format using a GCE modified with AuNPs, rGO, PtNPs and MoS_2_@Cu_2_O nanohybrid	Amperometry	HBS Ag/human serum samples	0.5 pg∙mL^−1^–200 ng∙mL^−1^	0.15 pg∙mL^−1^	97.8–101.7	ELISA	[130]
SPCE	AuNPs and rGO	Label-free format involving SPCE modified with AuNPs and rGO	EIS	Anti- HBcAg/human serum samples	3.91−125.00 ng∙mL^−1^	3.80 ng∙mL^−1^	-	-	[131]
AuE	PTy-CNTs	Label-free format involving AuE modified with PTy-CNTs	SWV	Anti- HBcAg/blood samples	1.0–5.0 ng∙mL^−1^	0.89 ng∙mL^−1^	-	-	[132]
GCE	AuNPs	Sandwich format using GCE modified with AuNPs and PPy nanosheets as a sensing platform	Amperometry	HBS Ag/human serum samples	0.0005–10 ng∙mL^−1^	166 fg∙mL^−1^	97.6–101.3	-	[133]
AuSPE	ZnO nanorods and AuNPs	Label-free format using AuSPE modified with ZnO nanorods and AuNPs	CV	HCV/-	-	0.25 μg∙μL^−1^.	-	-	[134]
GCE	GQDS/AuNPs/PANI	Label-free format using GCE modified with GQDS/AuNPs/PANI	EIS	HEV/clinical samples	1 fg∙mL^−1^–100 pg∙mL^−1^	0.8 fg∙mL^−1^	-	RT-qPCR	[139]
SPCE	GO/Fe_3_O_4_/PB and AuNPs	Label-free format using SPCE modified with GO/Fe_3_O_4_/PB nanocomposites and AuNPs	DPV	HBS Ag/human serum samples	0.5 pg∙mL^−1^–200 ng∙mL^−1^	0.166 pg∙mL^−1^	-	ECLIA	[140]
SPCE	rGO/PB-Au@PtNPs	Label-free format using SPCE modified with rGO/PB-Au@PtNPs.	DPV	HBS Ag/human serum samples	0.25 pg∙mL^−1^–400 ng∙mL^−1^	0.08 pg∙mL^−1^	98.16–102.53	ECLIA	[141]
GCE	Cu-MOF	Label-free format using GCE modified with Cu-MOF nanospheres	DPV	HBS Ag/human serum samples	1–500 ng∙mL^−1^	730 pg∙mL^−1^	76.93–92.18	-	[143]
SPE	MNPs-pDA	Sandwich format ELIME using SPCE modified with MNPs-pDA	DPV	HAV/tap water	10^−10^–5 × 10^−7^ IU∙mL^−1^	1 × 10^−10^ IU∙mL^−1^	83	RT-qPCR	[144]
SPCE	ZnO-NWs	Label-free format using modified with ZnO-NWs	EIS	HIV p-24 Ag/-	-	0.14 pg∙mL^−1^	-	-	[149]
SPCE	AuNP−MNP−CNT CdSeTeS QDs	Sandwich format including SPCE modified with AuNP−MNP−CNT nanocomposite and CdSeTeS QDs	DPV	Influenza A/-	1 fg mL^−1^–1 μg∙mL^−1^	9.05 fg∙mL^−1^	-	-	[151]
ITOE	mPC	Label-free format using ITOE modified with nanostructured functionalized mPC via APTES	CV	SAA biomarker Influenza A H1N1/	30–70 μg∙mL^−1^	3.1 μg∙mL^−1^	-	-	[153]
DEP array electrodes	AuNPs	Competitive format using DEP array electrodes modified with AuNPs	SWV	Recombinant Spike protein S1 as biomarker MERS COV/biological fluids samples	0.001–100 ng∙mL^−1^	1.04 pg∙mL^−1^	95–108	RT-PCR	[156]
DEP array electrodes	AuNps	Competitive format using DEP array electrodes modified with AuNPs	SWV	HCOV antigen/biological fluids samples	0.01–10,000 ng∙mL^−1^	0.4 pg∙mL^−1^	89–94	RT-PCR	[156]

Abbreviations: ab: antibody; APTES: 3-aminopropyl triethoxy silane; AIV: Avian Influenza Virus; AuE: gold electrode; AuSPE: gold screen printed electrode; Au/MNP-CNT: gold (Au)/iron-oxide magnetic nanoparticles-decorated carbon nanotubes; AuNPs: gold nanoparticles; AuNCs: gold nanoclusters; CNFs: carbon nanofibers; DPV: differential pulse voltammetry; ECLIA: electro-chemiluminescence immunoassay; EIM: enzyme-induced metallization; EIS: electrochemical impedance spectroscopy; ELIME: enzyme linked immunomagnetic electrochemical assay; ELISA: enzyme-linked immunosorbent assay; bi-FMNs: bifunctional fluorescence magnetic nanospheres; FMCNs: fluorescent-magnetic-catalytic nanospheres; GCE: glassy carbon electrode; GO: graphene oxide; rGO: reduced graphene oxide; GQDs: graphene quantum dots; LEG: laser engraved graphene; L.R.: linearity range; HBcAg: hepatitis B virus core antigen; HBS Ag: hepatitis B surface antigen; HCV: hepatitis C virus; HEV: hepatitis E virus; HIV: human immunodeficiency virus; ITO: indium doped tin oxide; LFIA: lateral flow immunoassay; LSV: linear sweep voltammetry; MA: microelectrode array; MNPs-pDA: Fe_3_O_4_@poly(dopamine) magnetic nanoparticles; MOF: metal–organic framework; (MoS_2_@Cu_2_O: molybdenum disulfide@cuprous oxide; mPC: mesoporous carbon; MWCNTs: multi-walled carbon nanotubes; NP: nucleocapsid protein; NSP: non-structural protein; PANI: poly(aniline); PB: Prussian Blue; PTy: poly(tyramine); QDs: quantum dots; RT-PCR: real time polymerase chain reaction-reverse transcription; RT-qPCR: real time quantitative polymerase chain reaction-reverse transcription; rGO: reduced graphene oxide; SAA: serum amyloid A; SARS-CoV-2: severe acute respiratory coronavirus syndrome; SCX8: *p*-sulfocalix[8]arene; SME: single molecule electrochemistry; SP: spike protein; SPCE: screen-printed carbon electrode; SPMWCNTE: screen-printed multi walled carbon nanotube electrode; SWCNTs: single walled carbon nanotubes; SWV: square wave voltammetry; TB: toluidine blue; ZnO-NWs: ZnO nanowires.

### 4.3. Aptasensors

Aptamer-based biosensors, or more briefly aptasensors, can involve an aptamer as a bioreceptor (also named capturing aptamer/probe) or transducer (also named signal aptamer/probe) [157].

Aptamers are small single-stranded artificial nucleotides (DNA or RNA) with an outstanding capability to bind to their targets, including proteins, viruses, bacteria, other cells and small molecules such as amino acids, nucleotides, and antibiotics [46,154].

The secondary and tertiary structures of aptamers enable a non-covalent and specific bond with the corresponding targets via aptamer-target recognition, involving, for example, aromatic rings, π-π system stacking, van der Waals forces, electrostatic interactions or hydrogen bonding and creating particular 3D configurations, such as stem, loop, hairpin, or G-quadruplex structures. Aptamers are often compared to antibodies because of their specific bond with the target and are also indicated as chemical antibodies or artificial antibodies [48,157,158]. Aptamers are selected using the well-known Systematic Evolution of Ligands by Exponential enrichment (SELEX) process, where many combinatorial libraries of oligonucleotides are examined using a repetitive procedure of in vitro selection and amplification for their affinity to a specific analyte [158]. Consequently, the SELEX process provides an increase in selectivity, removing the aptamers able to bond structures similar to the target and allowing the selection of aptamers for non-immunogenic or toxic molecules [158]. Just a reminder, immunogenicity is the capability of an antigen or aptamer, for instance, to induce an immune response and is generally considered to be an undesirable physiological response.

Electrochemical aptasensors can be classified as the labeled type, where labels are covalently or non-covalently linked to aptamers, or the label-free type [48].

Labeled aptasensors include different aptasensor formats, such as the sandwich one. In this case, the target molecule is captured between two aptamers: the capture aptamer and the labeled aptamer. The first is immobilized on the electrode surface, and the latter is used for the detection. The recognition process includes the displacement of a complementary sequence, previously hybridized by the capture aptamer and the switch-on/off. The switch-on/off is correlated to an aptamer conformational change after the bond with the target molecule, so causing a decrease/increase in the distance of the label from the electrode surface. In this case, the aptamer is modified with a redox probe to be quantified and correlated with the concentration of the target.

The label-free approach is related to the modifications of the interfacial properties of the aptasensor surface after the aptamer-target interaction. Consequently, the analyte is detected by recording changes in its electrochemical redox activity or in the electrochemical response of a redox probe [48].

Finally, analyzing the competitive approach, a competition takes place between a free oligonucleotide and the target for specifically binding an aptamer immobilized on the electrode [158].

The immobilization procedure is fundamental for establishing the aptasensors’ performances, guaranteeing proper reactivity, orientation, availability and stability of the immobilized aptamer and minimizing non-specific undesired adsorptions. The most diffused approaches are the covalent bond, the affinity reaction, and the self-assembled layer [159,160].

The immobilization is usually followed by incubation with appropriate blocking agents, as already mentioned in Section 4.2 for the immunosensors.

As a first example, we would like to describe the electrochemical label-free aptasensor for the detection of HBS Ag [161], using a GCE modified with a nanocomposite including AuNPs and rGO. Next, a thiol-terminated aptamer as BRE was covalently immobilized onto the nanocomposite through the thiol chemistry. Methylene Blue (MB), acting as an electrochemical probe, was inserted into the aptamer structure via its electrostatic interaction with the guanine bases. The working principle of an aptasensor depends on the specific binding between the aptamer and HBS Ag. In the absence of HBS Ag, a strong electrochemical signal was observed due to the MB as an intercalator. After the HBS Ag addition, MB was released from the sensing interface because HBsAg triggered an aptamer conformational change. The electrochemical response recorded by CV linearly decreased with the HBS Ag concentration increase in the range from 0.125 to 2.0 fg∙mL^−1^ with a limit of detection of 0.0014 fg∙mL^−1^.

PSA, vitamin C, glucose and fetal bovine serum (FBS) were tested as possible interferents, and the electrochemical response was not affected by the presence of these interfering molecules. The repeatability and reproducibility were also investigated with acceptable results in terms of RSD%: 4.6% (repeatability) and 5.3% (reproducibility). Finally, after three weeks at 4 °C, the electrochemical response decreased by only 7.7% with respect to its initial value. The aptasensor was then applied to spiked human serum samples with recoveries ranging from 90.4 to 104.15%.

A recent paper proposed a MERS-nanovesicles (NVs) aptasensor composed of multi-functional DNA aptamer and a hybrid nanocomposite including GO and encapsulated molybdenum disulfide (GO-MoS_2_), combining EIS and surface-enhanced Raman spectroscopy (SERS) as detection techniques [162]. MERS-NVs have a similar structure to the viral envelope but do not include genetic material [163]; they show higher stability and are not assumed as biohazardous. In addition, the MERS-NV aptamer was developed to bind specifically to the spike protein on MERS-NVs.

The multi-functional MERS aptamer (MF-aptamer) was realized by connecting the prepared aptamer to a DNA 3-way junction (3WJ) structure. DNA 3WJ can be described as a three arms structure, able to connect three systems, including a MERS aptamer (bioprobe), an MB (signal reporter), and a thiol group (linker). In addition, the GO-MoS_2_ nanocomposite acted to enhance the electron transfer and MoS_2_ to enhance the SERS response. Au micro-gap- electrode was employed as a working electrode to be modified with the multifunctional DNA aptamer and with the nanocomposite. A scheme of the aptasensor is reported in Figure 11.

MERS-NVs were detected with LOD of 0.176 pg∙ml^−1^ (SERS) and 0.405 pg∙ml^−1^ (EIS) in PBS buffer and with LOD of 0.525 pg∙ml^−1^ (SERS) and 0.645 pg∙ml^−1^ (EIS) in 10% diluted saliva. H5N1 AIV, H1N1 AIV, hemoglobin, amylase, and immunoglobulin G were used as interfering compounds, being present in saliva or being surface proteins of other respiratory viruses. They did not affect the electrochemical response of MERS-NVs. Unfortunately, the data on stability, reproducibility, and repeatability, together with those regarding the applicability to the real samples, were not provided.

Norovirus (NoV) is a well-known human foodborne pathogen triggering diarrhea and acute gastroenteritis in people of all ages worldwide. Human norovirus (HuNoV), already referred to as the “Norwalk virus” or “small round structured virus”, is an RNA virus belonging to the *Caliciviridae* family, which derives its name from the Greek word calyx (cup), by referring to cup-like depressions on the virus surface. Caliciviruses are small viruses of 30–35 nm in size, visible under a microscope as spherical particles without envelopes and spikes [164,165,166].

Noroviruses are divided into seven gene groups (from GI to GVII), including 30 genotypes. GI, GII and GIV are generally associated with human infections. In particular, considering the virus transmission, the GII.4 genotype is more commonly associated with transmission via direct person-to-person contact, while GI.3, GI.6, GI.7, GII.3, GII.6, and GII.12 are commonly transmitted via contaminated food products [164].

A sandwich-format electrochemical aptasensor using two recognition elements, i.e., Apt and specific peptide (HT), for HuNoV detection was recently reported [167]. In detail, the peptide was immobilized onto a GCE modified by Au NPs, via the Au–S coordination bond between Au and the thiol group of cysteine residues present in the peptide backbone. The aptamer was immobilized onto a magnetic nanocomposite Au@COF@Fe_3_O_4_, according to different steps. Firstly, the magnetic nanocomposite Fe_3_O_4_@COF was synthesized following a previous procedure [168] involving Fe_3_O_4_ nanoclusters and 1,3,5-triformylphloroglucinol (Tp) and 2,6-diaminoanthraquinone (DA), as COF starting materials. Then, AuNPs were incorporated in Fe_3_O_4_@COF, producing Au@COF@Fe_3_O_4_. Next, a macrocyclic molecule, i.e., a cationic pillar such as water-soluble pillar[5]arene (WP5A), selected for improving MB as an electrochemical indicator, was immobilized on Au@COF@Fe_3_O_4_ producing WP5A@Au@COF@Fe_3_O_4_ nanocomposite. Finally, Apt was immobilized onto WP5A@Au@COF@Fe_3_O_4_ via thiol functionalized AuNPs, and MB was anchored on the nanocomposite via WP5, so finally, the following bioconjugated MB@Apt@WP5A@Au@COF@Fe_3_O_4_ as a signal probe, was created.

Since the aptamer and specific peptide bind different domains of HuNoV capsid protein, in the presence of HuNoV, the sandwiching involved the signal probe and the peptide-modified GCE, and the redox probe MB was linked on the electrode.

After the optimization of the experimental condition, the HuNoV GII4 genotype was detected by means of DPV and a linear concentration range of 2.5–2.5 × 10^5^ copies∙mL^−1^ with a LOD of 0.8 copies∙mL^−1^ was achieved. Rotavirus (RV), Coxsackie virus (Cox), sucrose, glucose, BSA, and trypsin were tested as possible typical interferent molecules. They showed negligible changes in the DPV response of HuNoV.

The aptasensor stability was evaluated by measuring its electrochemical response after storage at 4 °C for 8 weeks, and the current response was not particularly affected.

Then, the sensor was applied to real samples of food (strawberries and oysters) and to fecal samples. The recoveries were 97.1–103.9%, 98.6–102.2%, and 99.2–102.8% for spiked oysters, strawberries, and fecal samples, respectively.

Finally, the real samples data were compared successfully with those coming from rapid immunechromatographic assay (RIA).

A 3D label-free aptasensor was proposed for the detection of norovirus, including a movable spherical working electrode modified with a phosphorene-gold nanocomposite (BPAuNCs) and screen-printed electrodes [169], as shown in Figure 12. Briefly, multiple layers of BP-AuNCs were synthesized through the in-situ reduction of HAuCl_4_ in the presence of phosphorene nanosheets (BPNSs) as a reducing agent. The BP-AuNCs provided a proper supporting platform for aptamer immobilization, accelerating the electron transfer and enhancing the electrochemical response. The binding between the aptamer and norovirus induced a change in charge transfer on electrodes, and it can be used to quantify the virus.

In fact, a significant decrease in the peak current is observed after the norovirus sample addition due to the increase of steric hindrance, indicating the formation of a complex between the aptamer and the target. Consequently, the current changes can be correlated to norovirus concentration.

A linear relationship between DPV responses and the logarithm of NoV concentrations in the range of 1 ng∙mL^−1^ to 10 μg∙mL^−1^ with a LOD of 0.28 ng∙mL^−1^ was obtained.

Concerning the aptasensor stability, after 30 days at 4 °C, a decrease of only 2.9% in the electrochemical response was observed. The repeatability and reproducibility were analyzed, and acceptable results were achieved in terms of RSD%: 2.29% (repeatability) and 3.03% (reproducibility).

Different types of viruses, such as RV and astrovirus, were used to address the selectivity issues, and their corresponding responses were more similar to the blank response, so evidencing the specific norovirus capability to interact and bind to the aptamer.

The aptasensor was then applied to spiked real samples of fresh Pacific oysters, and the recoveries ranging from 97 to 106% were evaluated as satisfactory.

A label-free electrochemical aptasensor for H5N1 AIV [170] was realized using an approach similar to that already described for MERS-nanovesicles (NVs) aptasensor composed of multi-functional DNA structure and a hybrid nanocomposite [162].

Briefly, AIV H5N1 can infect a human from a bird [105], and the LPAIVs of H5 and H7 subtypes can mutate in HPAIVs, so inducing severe hemorrhagic disease and creating serious problems for both public health and the poultry industry, as already reported in Section 4.2 [103,104,105,106].

The multi-functional DNA structure included DNA 3 way-junction (3WJ) acting as three arms system where hemagglutinin (HA) protein detection aptamer (recognition element), horseradish peroxidase (HRP)-mimicked DNAzyme (electrochemical signal reporter), and thiol group (linker for immobilizing the DNA 3WJ structure) are connected. Hemagglutinin (HA) protein is a biomarker for the detection of AIV H5N1.

It is well-known that the HRP-mimicked DNAzyme with the hemin can be considered a useful tool for simplifying the detection step [171] and for amplifying the electrochemical signal [172]. It required a cost-effective short oligonucleotide and performed similar redox properties to the HRP enzyme.

In addition, pAuNPs (porous AuNPs), cast on the Au electrode surface, were introduced to improve electron transfer. Then, the DNA 3WJ structure, including the aptamer, was covalently immobilized on the electrode surface via a thiol linker and the resulting DNA 3WJ/pAuNPs/Au electrode was used as a working electrode.

Considering the aptasensor analytical performances, HA protein was detected by means of CV and a LOD of 1 pM both in buffered solution and in diluted-chicken serum was obtained.

BSA, troponin I and myoglobin were used as interferent proteins and the HA protein electrochemical response was not affected by them. Unfortunately, the reproducibility, repeatability and stability data were not provided together with those related to the applicability to real samples.

A single-nanoparticle collision electrochemistry (SNCE) biosensor (SNCEB) based on the SNCE electrocatalytic mode for the detection of the H7N9 avian influenza virus (AIV) was recently reported [173]. It is to be underlined that SNCE is confirmed as a promising nanoparticle-based analytical method [174]. Briefly, a constant potential is applied to an ultramicroelectrode (UME), immersed in a dilute solution of NPs. Owing to the Brownian motions, stochastic collisions of the NPs occurred at the surface of the UME, causing transient electrochemical signals. [174] Considering the different ways of interaction between nanoparticles and the UME, the SCNE strategies can be divided into three conventional categories according to the electrochemical process: blocking, electrocatalytic, and particle self-electrolysis. In particular, in the electrocatalytic mode, [175,176,177] the catalytically active NPs are present in the solution, producing electrocatalytic processes at the UME surface. In this context, using specific BREs, such as antibodies and aptamers, the specificity of the method can be exploited and improved. On the other hand, it is to be evidenced that an efficient signal amplification strategy to perform a sensitive and specific detection of biological analyte is necessary for assembling a biosensor realistically exploiting the SNCE electrocatalytic mode.

Coming back to the SNCE-based aptasensor, nucleic acid aptamers acted as BREs for the target virus detection. Single-strand DNA1 (ssDNA1) was released after the recognition event and hybridized with another single-strand DNA2 (ssDNA2). After that, with the help of the nicking endonuclease enzyme (Nt.AlwI), Au NPs were released, and stochastic collisions of the AuNPs occurred, so it is possible to quantify the target virus amount. For improving the virus detection, nucleic acid aptamers were used, and the H7N9 AIV determination was transformed in nucleic acid detection. Also, magnetic nanospheres (MNs) were used to immobilize the H7N9 AIV aptamers. These aptamers hybridized with ssDNA1, so producing H7N9-apt-ssDNA1 conjugates. After the recognition process of H7N9 AIV through H7N9-apt-ssDNA1 conjugates, ssDNA1 was released and then hybridized with ssDNA2 to form a double-strand DNA (dsDNA) molecule.

ssDNA2 was connected with AuNPs and MNs (MNs-ssDNA2-AuNPs). Nt.AlwI can cleave ssDNA2 in the dsDNA, while ssDNA1 remained unaltered. On the other hand, AuNPs bound to ssDNA2 were cleaved and used for electrochemical detection. After the AuNPs collision at the UME surface and by counting the SNCE frequencies of Au NPs, the concentration of H7N9 AIV can be obtained indirectly by means of chronoamperometry. The detection principle of the SNCEB is illustrated in Figure 13.

After the optimization of the experimental conditions, a linear concentration range from 0.2 pg∙mL^−1^ to 200 ng∙mL^−1^ with a LOD of 24.3 fg∙mL^−1^ was determined.

The aptasensor selectivity was studied, including different viruses such as H9N2 AIV and the Japanese encephalitis virus (JEV). The collision frequency was almost negligible in the presence of H9N2 AIV and JEV, indicating a proper specificity of the SNCEB for H7N9 AIV detection. The matrix effect was analyzed using serum, and no significant differences were evidenced if the obtained data were compared with those obtained in a buffered solution. Unfortunately, the reproducibility, repeatability and stability data were not provided together with those related to the applicability to real samples.

Finally, I would like to introduce two examples of aptasensors for SARS-CoV-2 detection.

An electrochemical dual-aptamer biosensor based on the MOFs NH_2_-MIL-53(Al) decorated with Au@Pt nanoparticles and the following enzymes such as HRP and hemin/G- quadruplex DNAzyme (GQH DNAzyme) as signal nanoprobe was reported for the detection of SARS-CoV-2 nucleocapsid protein (NP) [178]. MOFs NH_2_-MIL-53 (Al) were synthesized with a hydrothermal approach using aluminum chloride hexahydrate and 2-amino-1,4-benzenedicarboxylic acid (NH_2_·H_2_BDC). Then, Au@Pt nanoparticles were incorporated into the MOFs structure via –NH_2_ moieties, and the resulting Au@Pt/NH_2_-MIL-53 was modified with HRP and with the thiolated aptamers (SH-2G-N48 and SH-2G-N61) including double G-quadruplex sequence, obtaining signal nanoprobes for amplifying the aptasensor response and able to co-catalyze the oxidation of hydroquinone (HQ) in the presence of H_2_O_2_. In addition, the two thiolated DNA aptamers (N48 and N61) were also immobilized on the AuE surface via gold-thiol interactions.

In the presence of SARS-CoV-2 NP, the sandwiching was realized by dropping the nanoprobes onto the AuE surface modified with the thiolated aptamers, and HQ was determined by means of DPV. The electrochemical responses showed a linear correlation to the logarithm of the protein concentration in the range from 0.025 to 50 ng∙mL^−1^ with a LOD of 8.33 pg∙mL^−1^.

Several proteins, such as the cTnI protein, Her2 protein, and MPT64 protein, were tested as possible interfering molecules, and a DPV response was obtained only in the presence of SARS-CoV-2 NP. The repeatability was investigated with acceptable results in terms of RSD%, ranging from 2.6 to 5.0%, depending on the SARS-CoV-2 NP concentration analyzed.

The matrix effect was investigated, and different concentrations of SARS-CoV-2 NP were analyzed. The results were comparable to those coming from the ELISA method, and the corresponding recoveries were in the range of 92.0–110%.

The last example included an aptasensor based on an SPCE modified with AuNPs where was immobilized an aptamer targeting the binding domain of the spike (S) protein of SARS-CoV-2 [179]. A scheme of the aptasensor assembling is reported in Figure 14.

The detection of SARS-CoV-2 S-protein was performed by means of EIS, yielding a linear relationship between the EIS responses and the S-protein concentration in a semi-logarithmic plot in the range from 10 pM to 25 nM with a LOD of 1.30 pM (66 pg∙mL^−1^). The selectivity of the aptasensor was investigated using S-proteins from SARS-CoV, MERS-CoV and SARS-CoV-2.

While the response of the aptasensor towards MERS S-protein was only negligible, the response towards S protein of the SARS-CoV was not negligible, around more than half respect to that of SARS-CoV-2 S-protein, probably because the S-proteins of SARS-CoV and SARS-CoV-2 have very similar sequences.

Concerning the stability, after three weeks at 4 °C, a decrease in the electrochemical response of only 15 was evidenced. Unfortunately, the reproducibility and repeatability data were not provided together with those related to the applicability to real samples.

As a general comment regarding the reported examples of aptasensors, we can observe that the LODs, independently of the analyte, achieved ng∙mL^−1^ or fg∙mL^−1^ in several examples. Concerning the format, generally, label-free is preferred.

It must be underlined that almost the aptasensors reported were not always applied to real samples and validated with reference analysis. This represents a drawback if the purpose is to introduce these sensors in real life.

The analytical performances of the reported aptasensors for the determination of viruses as well as the corresponding sensor formats, are summarized in Table 3.

## 5. Conclusions

In this section, I would like to summarize some considerations regarding the functional nanomaterials’ role, the electrodes involved, and the biosensors’ typologies. In addition, some comments concerning the analytical performances of the mentioned biosensors, their validation with an external method of analysis, and the multiple-analyte determining are introduced. Finally, critical issues, challenges, and future perspectives concerning biosensors for virus detection are discussed.

As a general comment, I would like to underline that perfect biosensors should be able to detect, in a complex matrix, concentrations of an analyte which depend on the kind of application field, i.e., food, clinical, pharmaceutical, using a very small volume and providing rapid, precise, and accurate results. In addition, they should be low cost, biocompatible, not toxic, and stable during the time and under whatever experimental environmental conditions.

The functional nanomaterials can support the resolution of these problems when applied to electrochemical biosensors, for example, enhancing the electrochemical response and assisting the immobilization procedure.

On the other hand, the nanomaterial should be stable for hours, days, or weeks under physiological conditions, in high ionic strength buffers, at room or body temperatures, in aqueous solutions and in the presence of air/oxygen. A proper surface modification developed for each new nanomaterial has to be designed to address all these critical issues, protecting and retaining, however, the nanomaterial’s functionality.

A nanomaterial can assist in the immobilization of the biorecognition element or can be used for the generation/amplification signal and/or for the analyte separation. In addition, proper functional groups available on its surface allowed the BRE’s immobilization, using different approaches such as NHS/EDC chemistry, glutaraldehyde, thiol chemistry, and so on.

Going towards a low-cost biosensor, the synthesis, functionalization, and integration of nanomaterials should be inexpensive and optimized, avoiding the usual variability between lots and hopefully involving a large-scale production to move the biosensors from the lab to a commercial market.

The analytical parameters of different biosensors, such as LOD, linearity range, and reproducibility, can be assumed as the correct indicators of how a functional nanomaterial can improve the performance of a biosensor, including a reduction in the duration of the analytical determination, which in the case of a virus was found to be a fundamental requirement, very often, however, data related to the stability of the nanomaterial in experimental conditions such as physiological ones are missing or of poor quality.

A comparison with the corresponding electrochemical biosensors for virus detection, reported in the literature and not supported by the nanomaterials, evidenced that the electrochemical biosensors, including nanomaterials/nanostructures, showed lower detection limits, larger linearity range of concentrations, faster response time, higher selectivity and higher sensitivity. All these enhancements in the analytical performances are probably due to the peculiar and already described properties of the nanoscale materials (see Section 3).

Finally, I would like to point out that considering the classification of nanomaterials reported in Section 3, AuNPs turn out to be the most used nanomaterial (see Table 1, Table 2 and Table 3), but, in general, the most used approach is to integrate the various nanomaterials in a nanocomposite. Nanocomposites and/or nanohybrids represented the best option, including nanoparticles, nanotubes, nanoclusters, quantum dots or polymers, and the corresponding nanostructures were very complex with tailored and tuned architecture (see Table 1, Table 2 and Table 3). The combination of different materials, improving the electron transfer capability, the conductivity, and the electrocatalytic properties of the starting materials can enhance the analytical performance of the sensors.

Summarizing, the following challenges have to be addressed for an effective application of functional nanomaterials to the biosensing area: optimizing their synthesis, investigating their long-term stability under physiological conditions and properly integrating them with biosensors and miniaturized analytical systems. Their toxicity, biodegradability and biocompatibility assessment should be mandatory since biosensors are designed as disposable devices or are even used in vivo, especially considering those for virus detection or, in general, for applications of clinical interest [180].

Considering the electrode typologies reported in this review (see Table 1, Table 2 and Table 3), I would like to underline that different types of electrodes are employed for virus detection ranging from the more conventional bulk electrodes, such as GCE, AuE, and ITOE, to SPEs.

In particular, GCE was the best option among the conventional bulk electrode, probably owing to its peculiar properties such as large window potential, easy functionalization and regenerability, as already mentioned in Section 2.2.

Screen-printed electrodes deserve special mention. In fact, screen-printed sensors appear as simple, easy-to-use, and disposable tools. Most of the biosensors mentioned in this review employed carbon-based SPEs, involving graphite, CNTs, or graphene as the carbon material. SPEs can be considered as an alternative choice with respect to the more conventional bulk electrodes because very small solution volumes are required for the analysis [35]. Screen-printed electrodes can be tuned for several application fields ranging from food analysis to medical diagnostics, so including virus detection.

It is to be evidenced that electrochemical biosensors based on screen-printed electrodes can offer an alternative regarding-the on-site analysis due to progress in nanomaterials and in biotechnology. As a consequence of the introduction of more sensitive biomarkers for clinical analysis, a transition from traditional lab-based techniques to miniaturized, cost-effective and quicker tests for health monitoring is made possible through SPEs.

In this review, some interesting examples, including paper-based electrochemical biosensing platforms, are reported. The interest in papers regarding biosensing applications can be explained by their low cost, fine thickness, flexibility, compatibility with different patterning methods, disposability, easy functionalization with proper groups, and finally, its preserving action of the activity of the biomolecules, viruses or biomarkers included or immobilized on them.

At this point, some comments on the different biosensor typologies are required. In this review, several examples of biosensors for virus detection are reported, but most of them are immunosensors. Apparently, the use of aptamers can overcome the constraints of immunosensors, such as size, stability, variability between batches, cost, and chemical modifications. In fact, they can be designed to sustain binding-induced conformational changes, so improving both the sensitivity and the selectivity of the biosensor.

For example, the aptasensors lifetime is significantly longer compared to immunosensors. On the other hand, immunosensor storage is usually shorter, only a few days, under refrigeration and wet conditions.

Again, regeneration of aptasensors can be easily achieved, while for immunosensors, regeneration is more difficult because the antibodies can result damaged irreversibly even if the bound target is released from the immunosensor, so the immunosensor’s regeneration is strictly dependent on the specific antibody and its durability.

Nevertheless, despite the advantages of aptamers, antibodies are still the best option for a capture probe in biosensors, probably because of the well-established use of antibodies as capture probes. Finally, it should be mentioned that aptamers, being nucleotides, seem to have less diversified functionalities than antibodies, being proteins.

Considering the genosensor typology, the number of the examples reported in this review is comparable to that of aptasensors if the data in Table 1 and Table 3 are considered. As already reported in the literature, sample preparation and reagent handling are still the main drawbacks to address towards up-to-date virus detection, in addition to requiring skilled personnel and sophisticated laboratory instrumentations.

Another important issue is the capability of detecting different analytes using a biosensing platform, as this is an attractive factor for commercial exploitation. Very few examples are available, and in particular, some of them regarded the detection of different viruses using a genosensing platform ad hoc designed for different viruses analysis [89] or a common immunosensing platform, or the detection of the same virus, but using different virus biomarkers, for instance in the case of SARS-CoV-2 spike protein or nucleocapsid protein.

Again, it should be noted that the examples are few and concern mainly the immunosensing approach, probably because it is the best studied and analyzed.

The analytical performance of the biosensors in terms of the linearity ranges, and the LOD results were very promising, generally achieving ng∙mL^−1^ and sometimes ag∙mL^−1^, notwithstanding the type of biosensor, its format, and the target virus and/or related biomarker.

Sensor selectivity was not always addressed properly, and in addition, the criterion of choice among potentially interfering molecules is not always clear; for instance, if interfering viruses were selected because they have similar sequences or are present in the same complex matrix to be analyzed. The criterion of interfering molecules selection is important if selectivity data for the same target analyte coming from different biosensors have to be compared.

Moreover, the biosensors’ reproducibility, repeatability, and stability data were not always considered; consequently, it is not always possible to compare the analytical performances, even if considering the same target.

It is to be underlined that the biosensors reported in this review were not always applied to real samples, but it is fundamental, especially in everything related to health and clinical diagnosis.

Validation with a standard method is required for accurately evaluating the biosensors’ analytical performances. The biosensors described in this review and validated with a reference analysis method (see Table 1, Table 2 and Table 3), how analytical performances comparable to those obtained with virus-detection reference methods such as PCR, LFIA; ECLIA, RIA and ELISA, evidencing improvements in many different aspects. For example, miniaturization is fundamental, allowing the construction of portable, versatile, and low-sample consumption devices. Moreover, the miniaturization of the electrochemical devices can help to extend the use of biosensors on site, providing rapid medical responses without sample transportation and complex treatment, so improving simplified analytical protocols for clinical diagnosis.

On the other hand, electrochemical biosensors are not currently commercially available except for the well-known glucometer.

Different criticalities have to be considered and evidenced, such as the stability of the biorecognition element layer and the related immobilization protocol, the complex assembling biosensor procedure involving high-cost materials and the sample preparation steps. Indeed, another important challenge relies on providing portable and reusable devices able to discriminate among different viruses with adequate specificity and sensitivity.

Finally, industrial interest in the commercialization of electrochemical biosensors is a critical issue since it is connected to market demand, considering all the costs associated with the new technologies introduction.

These are still great challenges to be addressed before electrochemical biosensors are commercially available devices.

In conclusion, although the industrial production and application of biosensors for viral detection evidence many problems to be solved, these devices seem to be promising for the real-time and continuous monitoring of specific pathogens. Furthermore, these devices are and will be extremely important in endemic and pandemic scenarios, ensuring a sensitive and specific detection of viruses/pathogens.

## Figures and Tables

**Figure 1 molecules-28-03777-f001:**

Layout of an electrochemical biosensor for virus detection, including analytes, BREs, transducers and electroanalytical techniques.

**Figure 2 molecules-28-03777-f002:**
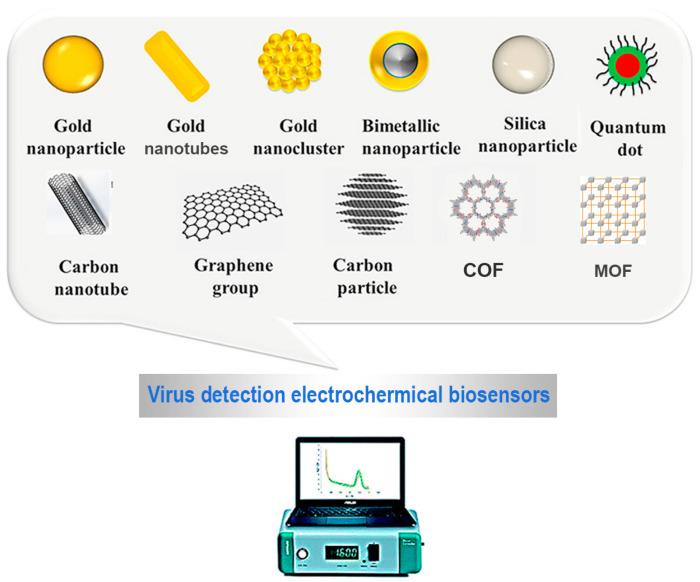
Schematic representation of the nanomaterials applied to electrochemical biosensors for virus detection.

**Figure 3 molecules-28-03777-f003:**
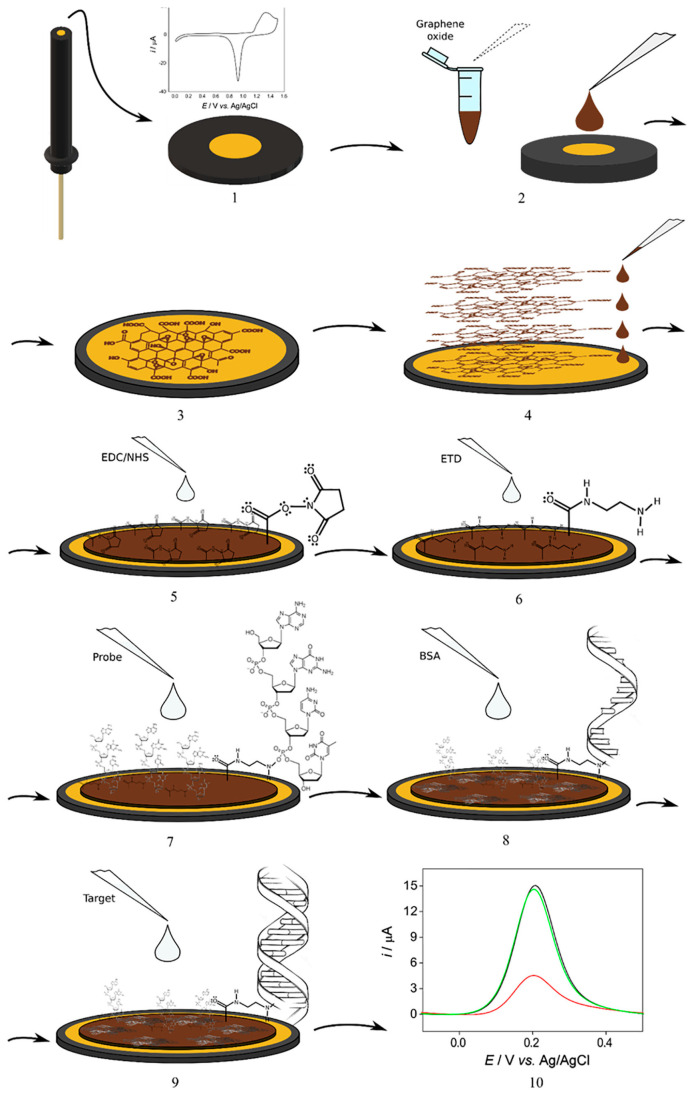
Schematic illustration of the assembling steps of the described genosensor. (1) electrode preconditioning; (2) drop-casting of GO on the AuE; (3) and (4) GO layers on Au surface; (5) surface activation with EDC/NHS; (6) introduction of (ethylenediamine) ETD; (7) probe immobilization; (8) surface blocking step; (9) target-probe interaction; (10) electrochemical detection, where: black– probe, red -positive control, green—negative control. Reprinted with permission from [86]. Copyright 2019, Elsevier.

**Figure 4 molecules-28-03777-f004:**
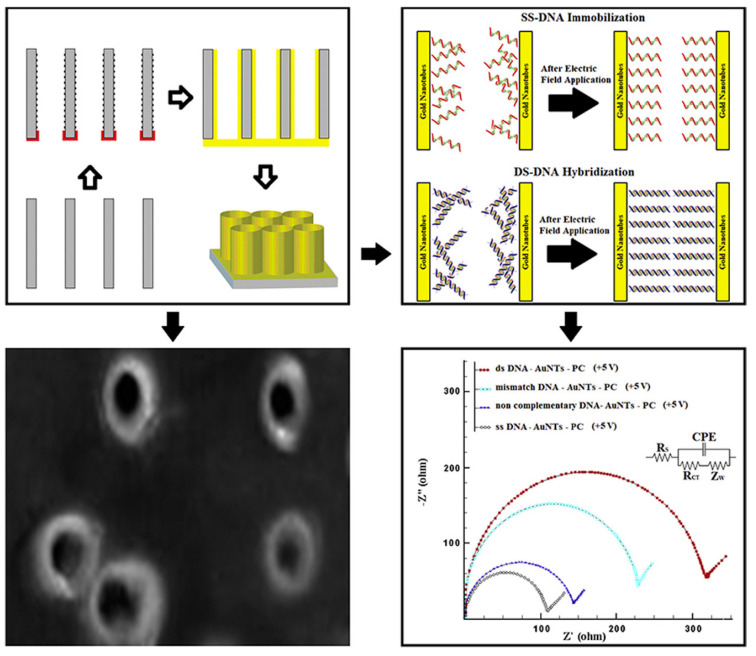
Scheme of the genosensor assembly and HPV detection steps. Reprinted with permission from [89]. Copyright 2019, Elsevier.

**Figure 5 molecules-28-03777-f005:**
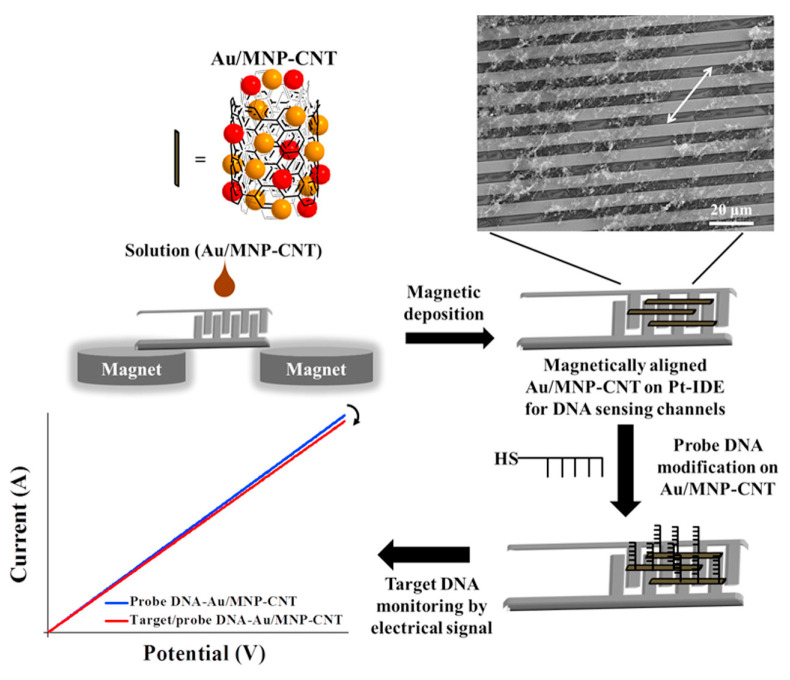
Scheme of the preparation of the magnetically aligned Au/MNP-CNTs on the Pt-IDE for DNA sensing channels (not to scale) and the genosensing mechanism involving the voltammetric determination of Influenza A virus. Reprinted with permission from [92]. Copyright 2018, Elsevier.

**Figure 6 molecules-28-03777-f006:**
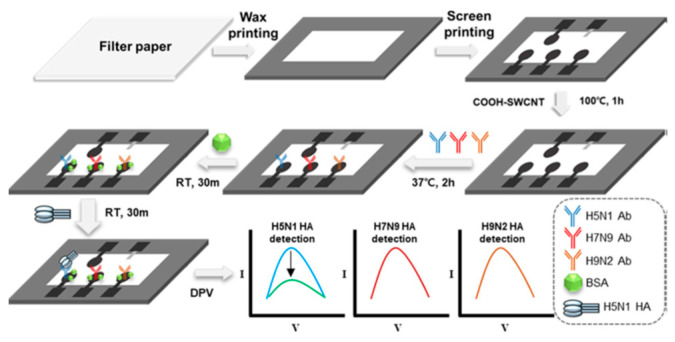
Scheme of the electrochemical immunosensor for detection of three different avian influenza virus antigens. Here, H5N1 AIV is only present for reasons of synthesis. RT indicates room temperature, ~25 °C. Reprinted from [112].

**Figure 7 molecules-28-03777-f007:**
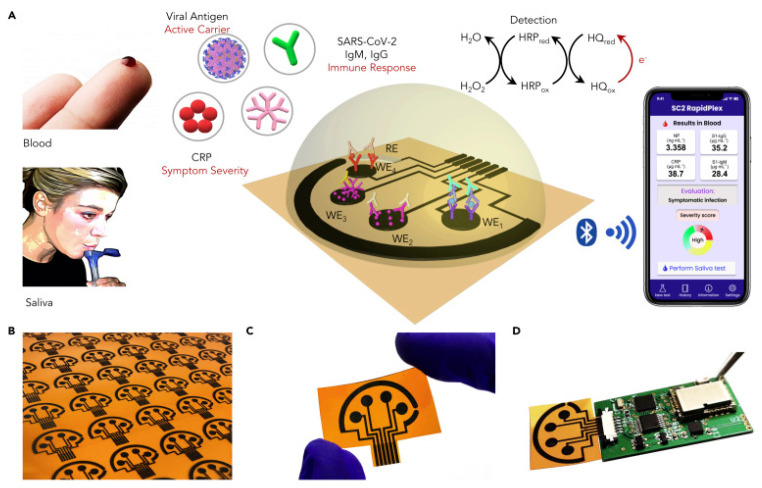
(**A**) Scheme of the SARS-CoV-2 RapidPlex multisensing telemedicine platform for detection of SARS-CoV-2 viral proteins, IgG and IgM, and/or CRP. (**B**) Mass-producible laser-engraved graphene sensor arrays. (**C**) Image of a disposable and flexible graphene array. (**D**) Image of a SARS-CoV-2 RapidPlex system with a graphene sensor array connected to a printed circuit board for signal processing and wireless communication. Reprinted with permission from [113]. Copyright 2020, Elsevier.

**Figure 8 molecules-28-03777-f008:**
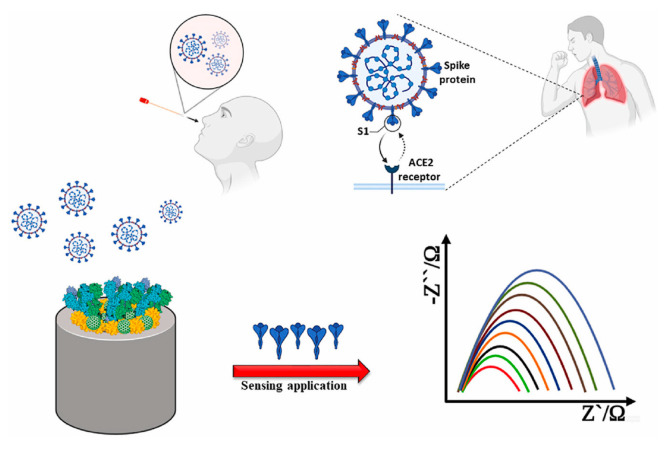
Illustration of the assembling sensor and the immunosensing mechanism. Reprinted with permission from [119]. Copyright 2022, Elsevier.

**Figure 9 molecules-28-03777-f009:**
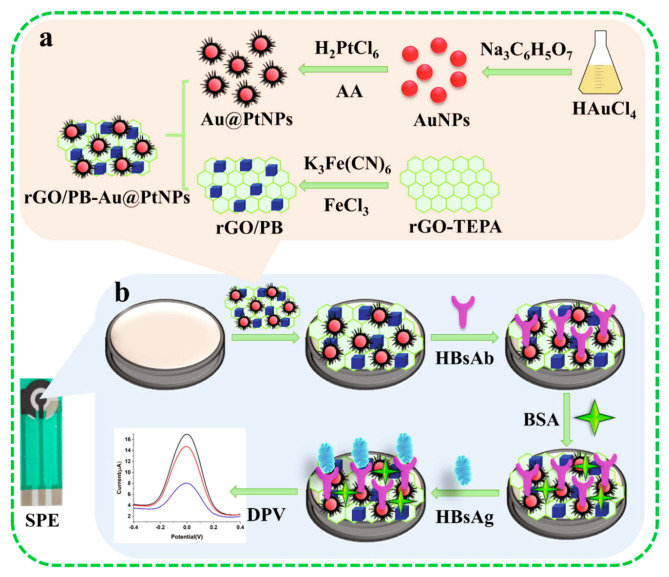
Illustration of (**a**) the nanocomposite synthesis and (**b**) sensor assembling and the immunosensing mechanism. Reprinted with permission from [141]. Copyright 2021, Elsevier.

**Figure 10 molecules-28-03777-f010:**
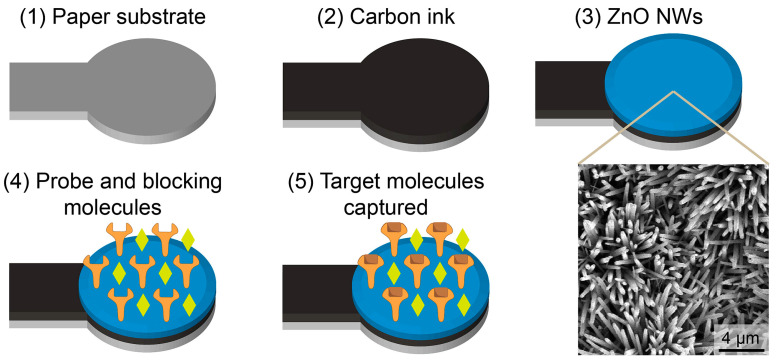
Preparation and surface functionalization of a paper-based working electrode. Five steps are involved: (1) cutting paper pieces into the working electrode shape, (2) printing a layer of carbon ink on paper, (3) growing ZnO NWs on the carbon ink, (4) immobilizing probe and blocking molecules to the surface of WEs, and (5) using the WE to capture target molecules for EIS biosensing. Reprinted with permission from [149]. Copyright 2021, Elsevier.

**Figure 11 molecules-28-03777-f011:**
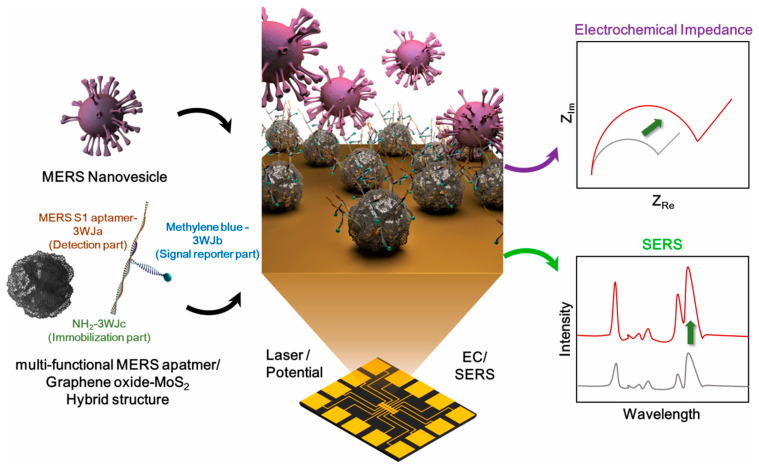
Scheme of the MERS-NV aptasensor based on EC/SERS dual approach. Reprinted with permission from [162]. Copyright 2022, Elsevier.

**Figure 12 molecules-28-03777-f012:**
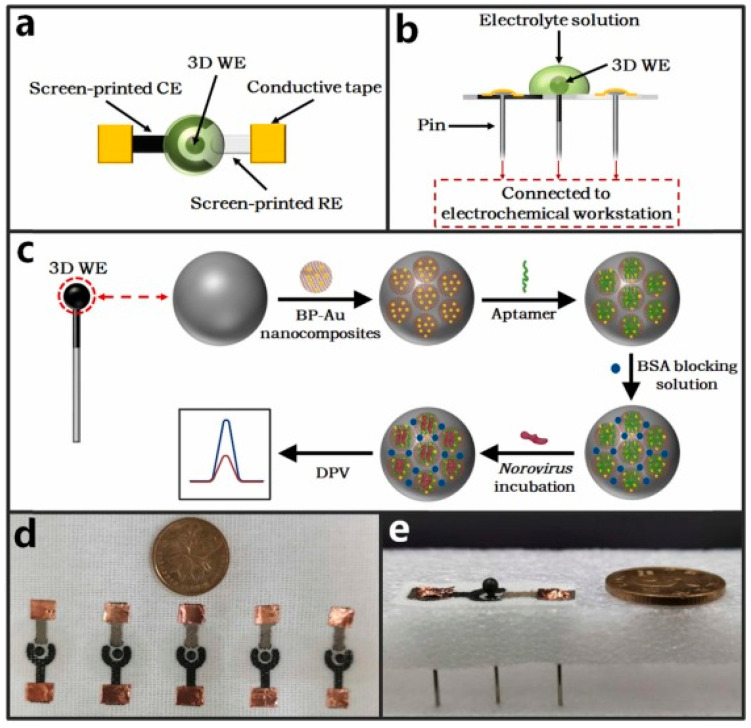
Scheme of the 3D electrochemical aptasensor for HuNoV detection. (**a**) Screen-printed CE and RE, and spherical WE. (**b**) Side-view of the proposed aptasensor and electrical connection. (**c**) Steps for the electrode functionalization and aptasensing of norovirus. (**d**,**e**) Prototype of the 3D electrochemical aptasensor. Reprinted with permission from [169]. Copyright 2022, Elsevier.

**Figure 13 molecules-28-03777-f013:**
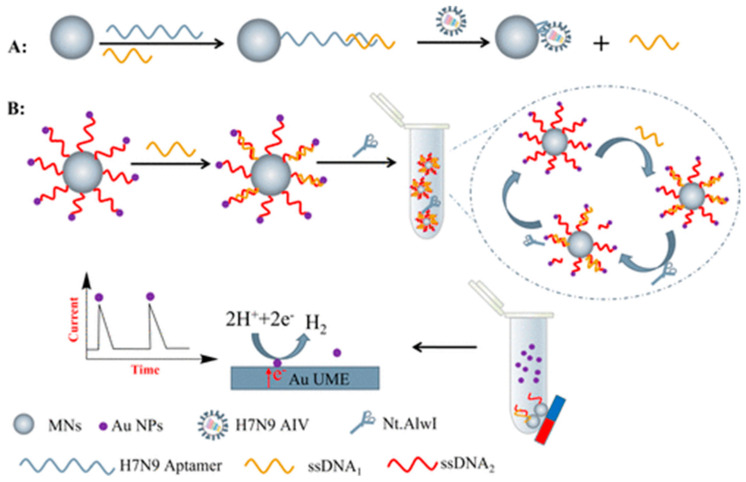
Representation of the SNCEB for the Detection of H7N9 AIV. (**A**) Modification of MNs with the aptamer, aaDNA1and interaction with virus; (**B**) MNs modified with AuNPs, ssDNA2, interaction with ssDNA1 and Nt.Alwl, and detection of AuNPs. Reprinted with permission from [173]. Copyright 2022, American Chemical Society.

**Figure 14 molecules-28-03777-f014:**
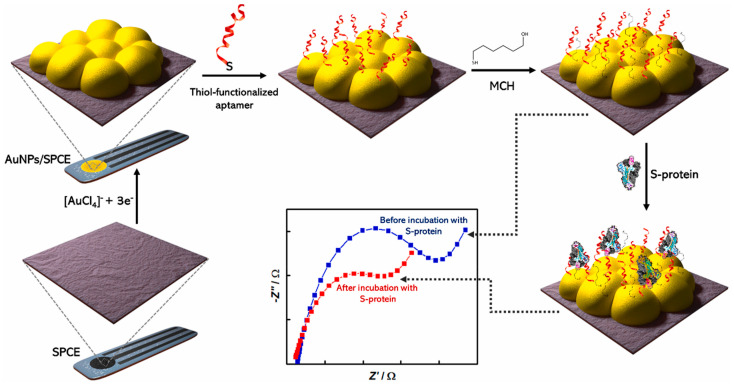
Preparation step by step of aptasensor for SARS-CoV-2 S-protein detection. Reprinted with permission from [179]. Copyright 2022, Elsevier.

**Table 1 molecules-28-03777-t001:** Performances of electrochemical genosensors for the viruses’ detection.

Electrode	Functional Nanomaterial	Genosensor Format	Electrochemical Technique	Analyte/Sample	L. R.	LOD	Recovery %	Reference Method	References
AuμE	G and AuNPs	Genosensor including a paper-based sensing platform with G and AuNPs	Non-conventional electrochemical technique collecting current-voltage data	SARS-CoV-2/Samples of nasopharyngeal swabs	585.4–5.854 × 10^7^ copies/μL.	6.9 copies/μL	-	FDA-approved RT-PCR COVID-19 diagnostic kit	[71]
DEP-Chips	GONC	Genosensor including GONC	DPV	SARS-CoV-2/-	1 × 10^−10^–1 × 10^−5^ mol∙L^−1^	186 × 10^−9^ mol∙L^−1^	-	-	[74]
GCE	GQDs	Label-free genosensor including GQDs	DPV	HBV/-	50–100 nM	1 nM	-	-	[80]
AuE	rGO	Genosensor including rGO	DPV	HBV/-	-	7.65 ng∙μL^−1^.		-	[83]
FTOE	MB@SiNPs	Genosensor including MB@SiNPs	EIS	HCV/clinical samples	100–10^6^ copies/mL	90 copies/mL	-	-	[84]
AuE	rGO	Genosensor including rGO	DPV	HCV/serum samples	-	1.36 nmol∙L^−1^			[86]
AuNTs PC	AuNTs	Genosensor including AuNTs	EIS	HPV/-	0.01 pm–1μM	1 fM	-	-	[89]
GCE	NH_2_-IL-rGO/MWCNTs	Genosensor using NH_2_-IL-rGO/MWCNTs nanocomposite	DPV	HPV/clinical samples	8.5 nM–10.7μM	1.3 nM	94.0–102.5	-	[90]
Pt-IDE	Au/MNP-CNT	Genosensor using (Au/MNP-CNT) nanocomposite	LSV	Influenza/-	1 pM–10 nM	8.4 pM	-	-	[92]
Pt-IDE	Au/MNP-CNT	Genosensor using Au/MNP-CNT nanocomposite	LSV	Norovirus/-	1 pM–10 nM	8.8 pM	-	-	[92]
PGE	rGO/AuNPs	Genosensor using rGO/AuNPs nanocomposite	DPV	VHSV/	10^−10^–10^−4^ mol∙L^−1^	1.25 pM	-	-	[99]

Abbreviations: AuE: gold electrode; AuμE: gold microelectrode; Au/MNP-CNT: gold (Au)/iron-oxide magnetic nanoparticles-decorated carbon nanotubes; AuNPs: gold nanoparticles; AuNTs: gold nanotubes; DPV: differential pulse voltammetry; DEP-chips: disposable electrical printed chips; EIS: electrochemical impedance spectroscopy; FTO: fluorine-doped tin oxide; GCE: glassy carbon electrode; GO: graphene oxide; GONCs: graphene oxide nanocolloids; rGO: reduced graphene oxide; HBV: Hepatitis B; HCV: Hepatitis C; HPV: Human papillomavirus; IL: ionic liquid; L.R.: linearity range; LSV: linear sweep voltammetry MB@SiNPs: methylene blue (MB) doped silica nanoparticles; MWCNTs: multi-walled carbon nanotubes; NH_2_-IL-rGO: amine-ionic liquid functionalized reduced graphene oxide; PC: polycarbonate; PGE: pyrolytic graphite electrode; Pt-IDE: Pt-interdigitated electrode; SARS-CoV-2: severe acute respiratory coronavirus syndrome; VHS: Viral hemorrhagic septicemia.

**Table 3 molecules-28-03777-t003:** Performances of electrochemical aptasensors for the viruses’ detection.

Electrode	Functional Nanomaterial	Aptasensor Format	Electrochemical Technique	Analyte/Sample	L. R.	LOD	Recovery %	Reference Method	Ref.
GCE	rGO-AuNPs	Label-free format using a GCE modified with rGO-AuNPs nanocomposite and MB as an electrochemical indicator	CV	HBS Ag/human serum	0.125–2.0 fg∙mL^−1^	0.0014 fg∙mL^−1^	90.4–104.15	-	[161]
AuE	GO-MoS_2_	Label-free format including an AuE modified with GO-MoS_2_ nanocomposite and DNA 3WJ multifunctional structure	EIS	MERS-NVs/-	-	0.405 pg∙mL^−1^	-	-	[162]
GCE	AuNPs Au@COF@Fe_3_O_4_	Sandwich format using a GCE modified with AuNPs, MB as an electrochemical indicator and MB@Apt@WP5A@Au@COF@Fe_3_O_4_ as signal probe	DPV	HuNoV/strawberry, oysters, fecal samples	2.5–2.5 × 10^5^ copies∙mL^−1^	0.84 copies∙mL^−1^	97.1–103.9 (strawberry) 98.6–102.2 (oysters) 99.2–102.8 (fecal samples)	RIA	[167]
SPCE	BP-AuNCs	Label-free format, including an SPCE modified with BP-AuNCs	DPV	HuNoV/Pacific oysters	1 ng∙mL^−1^–10 μg∙mL^−1^	0.28 ng∙mL^−1^	97–106	-	[168]
AuE	AuNPs	Label-free format using an AuE modified with pAuNPs and DNA 3WJ multifunctional structure	CV	AIV H5N1/-		1 pM	-	-	[169]
Au UME	AuNPs	SNCE aptasensor based on the SNCE electrocatalytic mode using AuNPs and MNs	chronoamperometry	AIV H7N9/-	0.2 pg∙mL^−1^–200 ng∙mL^−1^	24.3 fg∙mL^−1^	-	-	[173]
AuE	Au@Pt/MIL-53 (Al)	Sandwich format using an AuE with two aptamers immobilized on it and Au@Pt/MIL-53 (Al) nanocomposite modified with HRP and hemin/G- quadruplex DNAzyme as signal nanoprobe	DPV	SARS-CoV-2 NP/	0.025–50 ng∙mL^−1^	8.33 pg∙mL^−1^	92–110	ELISA	[178]
SPCE	AuNPs	Label-free format based on an SPCE modified with AuNPs	EIS	SARS-CoV-2 SP/	10 pmol∙ L^−1^–25 nmol L^−1^	1.30 pmol∙L^−1^	-	-	[179]

Abbreviations: AIV: avian influenza virus; AuE: gold electrode; pAuNPs: porous gold nanoparticles; AuNPs: gold nanoparticles; AuNCs: gold nanoclusters; COF: covalent organic frameworks; DPV: differential pulse voltammetry; EIS: electrochemical impedance spectroscopy; ELISA: enzyme-linked immunosorbent assay; GCE: glassy carbon electrode; GO: graphene oxide; rGO: reduced graphene oxide; HBS Ag: hepatitis B surface antigen; HuNoV: human norovirus; L.R.: linearity range; MB: methylene blue; MOF: metal-organic framework; MWCNTs: multi-walled carbon nanotubes; NP: nucleocapsid protein; RIA: rapid immunochromatographic assay; SNCE: single-nanoparticle collision electrochemistry; SP: spike protein; SPCE: screen-printed carbon electrode; UME: ultramicroelectrode.

## Data Availability

Not applicable.

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
