# Peer review of "Functional Nanomaterials Enhancing Electrochemical Biosensors as Smart Tools for Detecting Infectious Viral Diseases"

_molecules, 2023, doi:10.3390/molecules28093777_

Round 1

Reviewer 1 Report

The submitted work “Functional nanomaterials enhancing electrochemical biosensors as smart tools for detecting infectious viral diseases“ is interesting and important for the biosensors field. The comments listed below must be addressed before the manuscript can be accepted for publication. After addressing those major comments, it could be considered for publication.

1. Generally, there are small sentences in a kind of separate paragraphs that have to be joined into a bigger paragraph.

2. COVID-19 is a disease, please make it clear in paragraph 34.

3. Figure 1. In analyte, the word “Influenza” needs to be corrected.

4. Something is missing in line 170. To obtain what?

5. Please elaborate more on the next sentence, why is that an advantage? and what real advantage brings that? please mention it.

Line 204: “It should be highlighted that the main advantage of using nanomaterials to assemble electrochemical biosensors are due to the integration of macroscopic systems such as bulk electrodes with nanostructured materials.”

6. Line 576: Electronic.

7. I suggest the author use “I” instead of “we” since as a single author this is confusing.

8. The reviewed manuscript requires a bit of structure, maybe could be better to classify by nanomaterials used since that is the scope of this work, or to join similarly used nanomaterials (e.g. AuNP) and to extract general information on how they enhance the signal detection.

9. A comparison should be done of how those reported sensors improved their performance by the use of nanomaterials compared with other reported sensor that does not use a nanomaterial e.g. how many orders of magnitude enhancement they provide, it needs to be more critically discussed. Furthermore, it should discuss if the LOD and Dynamic detection range covers the clinically relevant range for the detection of those viruses as for the real application. This is not discussed and it could be interesting for the reader.

10. Table 1, what is L.R.? Please list at the bottom of the table the meaning.

11. No references in the conclusion and please conclude based on all the previously reported about the performance of all those sensors by utilizing nanomaterials and if they really enhanced the detection e.g. you claimed “Nanocomposites and/or nanohybrids represented the best option”, elaborate your conclusion based on why are they the best option. Synthesize the conclusion with the main key aspects.

12. Please adjust the tables so that everything fits in, there are many cut words and it is not possible to see them so clearly (e.g. reducing font size). Otherwise, the important content in the table is lost.

Please check and correct a few misspellings.

Author Response

My reply to Reviewer 1 comments are present in the attached file

Reviewer 2 Report

This manuscript deals with "Functional nanomaterials enhancing electrochemical biosensors as smart tools for detecting infectious viral diseases" I suggest a minor correction and require a detailed clarification. A correction should be addressed by the authors as follows: The abstract is not well organized; the sentences are incomplete, and there is no sense of continuity. It would be feasible if you included the significance of the current study in the abstract. A brief description of how the authors selected information from the literature in the databases, as well as what time period they searched for, is missing. The authors should justify and expand the information on the advantages of electrochemical biosensors  for biomedical applications. Authors should specify the main experimental conditions used based on the evidence from the literature. Where they briefly describe the most important data reported in the literature in a homogeneous manner and reinforce the relevance of Persea americana (avocado) seed husk mediated hydronium jarosite nanoparticles as novel alternatives. Authors should discuss whether the use of electrochemical biosensors  represents a solid alternative to existing therapeutics. Please add the below studies to your manuscript in the discussion section and bold your study novelties:

-DOI: 10.1016/j.foodchem.2021.129763

-DOI: 10.2217/nnm-2020-0441

Author Response

My reply to the reviewer 2 comments are present in the attached file

Round 2

Reviewer 1 Report

The author addressed my comments and suggestions, thus, I agree that this review manuscript can be accepted for publication.